# A numerical sensitivity study on the snow darkening effect by black carbon deposition over the Arctic in spring

Zilu Zhang[1,2], Libo Zhou[1,2,3*], Meigen Zhang[2,3*]

[1] Department of Lower Atmosphere Observation Research (LAOR), Institute of Atmospheric Physics, Chinese Academy of Sciences, Beijing, China
[2] College of Earth and Planetary Science, University of Chinese Academy of Sciences, Beijing, China
[3] State Key Laboratory of Atmospheric Boundary Layer Physics and Atmospheric Chemistry (LAPC), Institute of Atmospheric Physics, Chinese Academy of Sciences, Beijing, China

*Correspondence to*: Libo Zhou (zhoulibo@mail.iap.ac.cn);Meigen Zhang (mgzhang@mail.iap.ac.cn)

**Abstract**

The rapid warming of the Arctic, driven by glacial and sea ice melt, poses significant challenges to Earth's climate, ecosystems, and economy. Recent evidence indicates that the snow-darkening effect (SDE), caused by black carbon (BC) deposition, plays a crucial role in accelerated warming. However, high-resolution simulations assessing the impacts from the properties of snowpack and land–atmosphere interactions on the changes in the surface energy balance of the Arctic caused by BC remain scarce. This study integrates the Snow, Ice, Aerosol, and Radiation (SNICAR) model with a polar-optimized version of the Weather Research and Forecasting model (Polar-WRF) to evaluate the impacts of snow melting and land–atmosphere interaction processes on the SDE due to BC deposition. The simulation results indicate that BC deposition can directly affect the surface energy balance by decreasing snow albedo and its corresponding radiative forcing (RF). On average, BC deposition at 50 ng g$^{-1}$ causes a daily average RF of 1.6 W m$^{-2}$ in offline simulations (without surface feedbacks) and 1.4 W m$^{-2}$ in online simulations (with surface feedbacks). The reduction in snow albedo induced by BC is strongly dependent on snow depth, with a significant linear relationship observed when snow depth is shallow. In regions with deep snowpack, such as Greenland, BC deposition leads to a 25–41% greater SDE impact and a 19–40% increase in snowmelt than in areas with shallow snow. Snowmelt and land–atmosphere interactions play significant roles in assessing changes in the surface energy balance caused by BC deposition based on a comparison of results from offline and online coupled simulations via Polar-WRF/Noah-MP and SNICAR. Offline simulations tend to overestimate SDE impacts by more than 50% because crucial surface feedback processes are excluded. This study underscores the importance of incorporating detailed physical processes in high-resolution models to improve our understanding of the role of the SDE in Arctic climate change.

## 1. Introduction

Arctic amplification (AA) refers to the unprecedented rate of near-surface warming over the Arctic, which strongly impacts the Earth's climate (AMAP, 2021; Li et al., 2020), ecosystems (Myers et al., 2020) and economy (Salzen et al., 2022). The physical mechanisms responsible for AA include local climate feedbacks in the Arctic (e.g., surface albedo feedback, water vapor feedback and Planck feedback) as well as poleward heat and moisture transport from lower latitudes (Previdi et al., 2021; You et al., 2021). Although there is currently no consensus on the dominant process for AA, snow albedo feedback is generally considered an important factor, especially during melting period (Bintanja et al., 2012; Bokhorst et al., 2016; Guo & Yang, 2022). Recently, a number of studies have shown that the deposition of light absorbing particles (LAPs, e.g., black carbon, dust, brown carbon) onto Arctic snow surfaces could greatly affect rapid warming in the Arctic by reducing snow albedo (Dou & Xiao, 2016; Flanner, 2013; Kang et al., 2020; Qian et al., 2014).

Black carbon (BC), primarily generated from incomplete combustion processes involving fossil fuels, biofuels, and biomass, stands out as the most efficient particulate species in the atmosphere in regard to absorbing visible light (Bond et al., 2013). As an important light-absorbing aerosol, BC can significantly influence the radiation balance through multiple mechanisms. In addition to the direct effect of absorbing or scattering solar radiation (Haywood & Shine, 1995), it can also

exert indirect and semidirect effects by modifying the distribution, lifetime, and microphysical attributes of clouds (AMAP, 2015; Dada et al., 2022). Notably, when BC is deposited onto snow and ice surfaces, it enhances the absorption of solar radiation, leading to greater atmospheric warming and subsequent melting of snow and ice (Clarke & Noone, 1985; Flanner et al., 2007; Quinn et al., 2011).

The highly reflective surface of snow and ice in the Arctic makes it a very sensitive region for BC deposition. The greater the BC deposition in snow and ice is, the lower the snow albedo will be, which accelerates snow melting and Arctic warming, and vice versa (Hansen & Nazarenko, 2004; Lau et al., 2018). Snow albedo is important in determining the surface energy budget of polar regions (e.g., Barry et al., 1993; Hall & Qu, 2006; Jacobson, 2004; Lamare et al., 2016). Darkened snow and ice could change the near-surface heat transfers in the Arctic, consequently influencing the Arctic climate. During the spring melt period, the relatively strong solar radiation combined with the near-maximum snowpack depth makes the impact of BC on the SDE particularly significant for the terrestrial Arctic surface (Doherty et al., 2010; Zhang et al., 2024). In addition, during this period, the aging and melting processes of snow also affect the reduction in snow albedo induced by BC (Dang et al.,2015; He & Ming, 2022). Despite its importance, the influence of snow processes on BC-induced changes in snow albedo, as well as their impact on the surface energy balance, has not been fully explored and need further investigation.

The physically based Snow, Ice, Aerosol, and Radiation (SNICAR) model  (Flanner et al., 2012b; Flanner & Zender, 2005) is widely used to estimate the contributions of BC to the reduction in snow albedo and its corresponding radiative forcing (RF). Flanner et al. (2007) incorporated SNICAR into the National Center for Atmospheric Research Community Atmosphere Model (NACR CAM3, Version 3) global climate model (GCM) to improve the quantification of climate forcing from BC in snow. Their results emphasized that snow darkening by BC in snow plays a significant role in the climate impact of carbon aerosols, especially in the Arctic region. Using the NCAR CAM5 coupled with SNICAR, Zhou et al. (2012) reported that in the spring season, the Arctic forcing increases from +0.29 W m$^{-2}$ to +0.37 W m$^{-2}$ due to BC deposition, they also suggested that BC snow forcing is sensitive to the wet deposition in the Arctic region. On the basis of field observations conducted by Doherty et al. (2010). Dang et al. (2017) used SNICAR to calculate the reduction in snow albedo caused by BC in the Arctic, and highlighted the impact of snowpack properties on the assessment of the SDE. SNICAR was also applied to quantify the reduction in snow and ice albedo caused by long-range-transported Asian dust (Zhao et al., 2022). Although many studies have been carried out, the key physical mechanisms and its impacts on the SDE resulting from BC are not well evaluated. Several studies have investigated the snow-darkening effect (SDE) induced by BC and other LAPs. Using the NASA GEOS-5 (National Aeronautics and Space Administration Goddard Earth Observing System, Version 5) climate model, Lau et al. (2018) examined the impacts of the SDE on regional surface energy and water balances over Eurasia from March to August. These authors suggested that the SDE can intensify the extreme hot days in summer. Huang et al. (2022) employed the WRF-Chem model to study the impurity-induced SDE in the Sierra Nevada during April–July. They found that the reductions in the snow water equivalent and snow depth induced by the SDE were 20 and 70 mm, respectively, in June; and emphasized the negative role of the SDE in the local ecosystem. Both studies underscore the significance of obtaining a deeper understanding of the impacts of the SDE caused by LAPs, especially at the regional scale. However, the impacts of the SDE due to LAPs deposition generally depend on several factors, including atmospheric processes and surface properties, therefore the SDE shows large spatial variability in different regions. As a result, accurately assessing the impacts of the SDE and the associated feedback mechanisms is challenging (Huang et al., 2022; Minder et al., 2016; Rohde et al., 2023).

Compared to the global climate model, weather models with high temporal and spatial resolutions have the potential to enhance our understanding of the short-term effects and feedback mechanisms of BC deposition on Arctic snow surfaces (Oaida et al., 2015; Rahimi et al., 2020; Rohde et al., 2023). For example, the reduced vertical and horizontal resolution of global models may fail to accurately represent the detailed atmospheric features created by high static stability in the Arctic (AMAP, 2021). Kang et al. (2020) emphasized the importance of enhancing the spatial resolution of models in order to more effectively capture the spatial variability of snowpack. Furthermore, the necessity for conducting simulations with high-

resolution models was also emphasized. To the best of our knowledge, no comprehensive studies have used weather models to investigate the impacts of the SDE resulting from BC deposition on Arctic surface energy balances.

In this study, the SNICAR model coupled with a polar-optimized version of the Weather Research and Forecasting model (Polar-WRF) was used to investigate the impacts of the SDE by BC deposition in spring time from online (with surface feedbacks) and offline (without surface feedbacks) simulation experiments. The questions that we addressed in this study are as follows: (a) How does BC deposition affect surface energy exchange in the Arctic? (b) What are the crucial physical processes affecting the impacts of the SDE by BC deposition?(c) How the key surface feedback processes affect the SDE caused by BC? The remainder of this paper is organized as follows: In Section 2, we explain the methodology and assumptions used in this study. This is followed by a validation of modeling performances in meteorological fields and surface energy balance (Section 3.1), an analysis of sensitivity tests between snow properties and albedo reduction (Section 3.2), a quantification of the spatial distribution of impacts of the SDE by given column-mean BC concentrations in snow (Section 3.3), a consideration of the temporal evolution of the SDE at different snow depth ranges (Section 3.4), and a comparison of physical mechanisms between off-line and on-line simulations (Section 3.5). We summarize the results and provide the conclusions in Section 4. The list of abbreviations is shown in the Appendix A4.

## 2. Methodology

In this study, the Polar-WRF version 4.1.1 was used to investigate surface exchange processes in the Arctic. The SNICAR model was employed to assess the reduction in snow albedo caused by BC. The impacts of the SDE induced by BC deposition on the Arctic surface were quantified by integrating SNICAR with Polar-WRF, both with and without surface feedbacks. The descriptions of the two models are provided in Section 2.1 and Section 2.2. The calculation of the surface energy balance is introduced in Section 2.3. The mixing ratio of BC in snow and relevant snow process are introduced in Section 2.4. The model configuration and descriptions of the observed data are detailed in Section 2.5. All the experiments conducted are summarized in Section 2.6.

### 2.1. Polar-WRF/Noah-MP

The Polar-WRF version 4.1.1 has been developed and optimized for use in polar climates by optimizing heat transfer processes through snow and ice and adding a comprehensive description of sea ice to the Noah and Noah-MP land surface models (LSM) (Bromwich et al., 2009; Hines & Bromwich, 2008; Hines et al., 2015). The key points include (a) optimize the treatment of heat transfer for ice sheets and revised surface energy balance calculation in the Noah and Noah-MP LSMs; (b) comprehensively describe sea ice in Noah and Noah-MP; and (c) improve cloud microphysics for polar regions. A detailed description of the Polar-WRF model is provided in Hines and Bromwich (2008) and Hines et al. (2015).

The main improvements of Polar-WRF are associated with the Noah and Noah-MP LSM. In modified Noah-MP, allow users to specify spatially varying sea ice thickness, snow depth on sea ice, and the sea ice albedo to optimize the treatment of heat transfer for sea ice. To improve the adaptability in the polar regions, the freezing point of seawater is set at 271.36 K, the surface roughness over sea ice and permanent land ice is set at 0.001 m, the snow emissivity is set at 0.98, the snow density over sea ice is set at 300 kg/m$^3$, the thermal conductivity of the transition layer between the atmosphere and snow is taken as the snow thermal conductivity, and whenever the upper snow layer exceeds 20 cm depth, it is treated as if the snow were 20 cm thick for heat calculation. Options also allow for an alternate calculation of surface temperature over snow surfaces or setting the thermal diffusivity of the top 0.1 m deep tundra soil to 0.25 W m$^{-1}$ K$^{-1}$, representative of highly organic soil. In addition, the droplet concentration in the Morrison 2-moment microphysics is reduced from 250 cm$^{-3}$ to 50 cm$^{-3}$, which is more applicable to polar regions. All improvements have enhanced the simulation capability of Polar-WRF in polar regions (Bromwich et al., 2013; Hines et al., 2015).

The community Noah land surface model with multiple parameterization options (Noah-MP) (Niu et al., 2011) was originally developed based on the Noah LSM (Chen et al., 1997; Ek et al., 2003) to improve its modeling capabilities with enhanced physical representations, This model includes a multilayer snowpack physics module, that incorporates several important features, such as liquid water storage and melt/refreeze capability. Noah-MP has been integrated as a land component in the WRF model and extensively utilized to investigate regional snow processes (Abolafia-Rosenzweig et al., 2022; Yang et al., 2021; You et al., 2023). Two options were implemented for snow surface albedo in Noah-MP: one adopted from CLASS (Verseghy, 1991) and the other from BATS (Yang et al., 1997). The equations of the two snow albedo schemes are listed in Appendices **A1** and **A2**. In CLASS, the snow albedo for both direct and diffuse radiation is the same, with a fresh snow albedo assumed to be 0.84. Snow aging is modeled as an exponential function of time, and the minimum value of snow albedo is 0.55. In BATS, the fresh snow albedo is 0.95 for the visible band and 0.65 for the near-infrared band. The aging process of snow is described as a function of time, ground temperature, and snow mass. However, neither of these schemes explicitly incorporates snow-aerosol-radiation interactions, rendering them unsuitable for assessing the impacts of the SDE by LAPs deposition (Oaida et al., 2015; Wang et al., 2020). Thus, in this study, the SNICAR model was integrated into Polar-WRF/Noah-MP, to assess the impacts of snowpack properties and land–atmosphere exchange on the reduction in surface snow albedo caused by BC deposition and the corresponding changes in the surface energy balance.

**2.2. SNICAR**

The SNICAR model is a multilayer two-stream model that accounts for vertically heterogeneous snow properties and the influence of underlying surface albedo, incoming solar radiation and the presences of LAPs (Flanner et al., 2021; Flanner et al., 2012b; Flanner & Zender, 2005). The model is based on the theory of Wiscombe and Warren (1980) and Warren and Wiscombe (1980), with the multilayer two-stream solution from Toon et al. (1989). SNICAR has been widely used to estimate snow albedo reduction and RF induced by LAPs deposition (Dang et al., 2017; Flanner et al., 2012a; Huang et al., 2022; Pedersen et al., 2015), and it has been coupled with several land surface models (LSMs), including the Community Land Model (CLM) within the Community Earth System Model (CESM) (Flanner et al., 2007; He et al., 2024),the DOE's Energy Exascale Earth System Model (E3SM) Land Model (ELM) (Hao et al., 2023) and the Simplified Simple Biosphere model version 3 (SSiB-3) within the Weather Research and Forecasting Model (WRF) (Oaida et al., 2015). Recently, it has also been integrated into the Noah-MP to enhance the snow radiative transfer process, and the results exhibited better performances than that of the default snow albedo scheme at validation sites (Lin et al., 2024).

The SNICAR model used in this study assumes that snowpack may contain the following nine LAPs: two BC aerosols (hydrophilic and hydrophobic), two OC aerosols (hydrophilic and hydrophobic), and five sand dust aerosols (particle sizes of 0.1-1.0, 1.0-2.5, 2.5-5.0, 5.0-10.0, and 10.0-100.0 µm). Each aerosol is associated with distinct optical properties, including the single scattering albedo, the mass extinction coefficient and the asymmetric scattering factor. These parameters are derived from look-up tables (Flanner et al., 2021; Flanner et al., 2012b). A detailed description of the computation of the optical properties of snow with aerosols in SNICAR can be found in Flanner et al. (2012b).

The SNICAR model requires inputs regarding the following environmental conditions: the solar zenith angle (SZA), downwelling spectral irradiance, and spectral albedo of the underlying surface (e.g., bare ground albedo). Information about the snow properties, including the snow depth, snow density, snow thickness for each layer and snow grain radius is also needed.  In this study, we assume that snow grains are spherical with radii of either 100 µm (new snow) or 1000 µm (old snow), which correspond to typical snow effective grain radii (Dang et al., 2017; Warren & Wiscombe, 1980). For BC in snow, we assume that all BC particles are uncoated and externally mixed with ice particles. Other inputs are provided by Polar-WRF/Noah-MP outputs. For on-line coupling simulation, input datasets that are required for SNICAR through an updated Model I/O interface. The vertical profile of BC in snow is set as the column mean for a given concentration.

Once the size of snow particles has been determined, SNICAR initiates the process of selecting the most appropriate optical parameters (single scattering albedo, mass extinction cross-section, and asymmetry scattering parameter). These parameters are based on the snow grain radius and the presence of aerosol particles in the snow, and are obtained by consulting look-up tables (Flanner et al., 2021). Other required inputs, such as the solar zenith angle (SZA), downwelling spectral irradiance, and spectral albedo of the underlying surface, are provided by Polar-WRF/Noah-MP. SNICAR then calculates the bulk snow albedo and the absorbed solar radiation flux in each snow layer through a series of computations. When the SZA is greater than 0° and a snow layer exists, SNICAR is called upon, and the snow albedo calculated by SNICAR is used to evaluate the impacts of the SDE by BC deposition on the Arctic surface.

## 2.3. Energy balance analysis

In this study, the surface energy balance is governed by

$$H_m + PH = HS + LH + SW_{down} - SW_{up} + LW_{down} - LW_{up} \tag{1}$$

where $H_m$ (W m$^{-2}$) is the net energy flux into the snow surface layer; and $PH$ (W m$^{-2}$) is the precipitation advected heat (0 in this study). SW (W m$^{-2}$), LW (W m$^{-2}$), LH (W m$^{-2}$), and HS (W m$^{-2}$) represent surface shortwave radiation, longwave radiation, latent heat flux, and sensible heat flux, respectively.

The HS and LH are computed based on the following bulk transfer relationships from Garratt (1992) :

$$HS = \rho_a \times C_h \times C_p \times U \times T_s - T_a \tag{2}$$

$$LH = \rho_a \times C_w \times C_{LH} \times U \times (q_s - q_a) \tag{3}$$

where $\rho_a$ (kg m$^{-3}$) is the air density, $C_p$ (J kg$^{-1}$ K$^{-1}$) is the air heat capacity, $C_{LH}$ (J kg$^{-1}$) is the specific latent heat of water vaporization, $U$ (m s$^{-1}$) is the 10 m wind speed. $T_s$ (K) and $T_a$ (K) are the temperatures at the surface and in the air, respectively. $q_s$ (kg kg$^{-1}$) and $q_a$ (kg kg$^{-1}$) are the specific humidities at the surface and in the air, respectively. $C_w$ and $C_h$ are the surface exchange coefficients for heat and moisture, respectively. They are assumed to be equal in this study and calculated based on the Monin-Obukhov (M-O) similarity theory (Brutsaert, 1982) :

$$C_h = \frac{\kappa^2}{\left[\ln\left(\frac{z - d_0}{z_{0m}}\right) - \psi_m\left(\frac{z - d_0}{L}\right)\right]\left[\ln\left(\frac{z - d_0}{z_{0h}}\right) - \psi_h\left(\frac{z - d_0}{L}\right)\right]} \tag{4}$$

where $\kappa$ is the von Karman constant, $z$ (m) is the reference height, $d_0$ (m) is the zero-displacement height; $L$ (m) is the M-O length; $\psi_m$ and $\psi_h$ are the stability functions for momentum and heat transfer, respectively. They are defined as in Chen et al. (1997). $z_{0m}$ (m) and $z_{0h}$ (m) are the surface roughness lengths for momentum and heat, respectively.

Furthermore, $q_s$ is computed as follows:

$$q_s = \frac{0.622 \times e_s(T_s) \times RH_s}{P_{sfc} - 0.378 \times e_s(T_s) \times RH_s} \tag{5}$$

where $e_s(T_s)$ (Pa) is the saturation water vapor pressure at the surface temperature ($T_s$ (K)); $RH_s$ is the surface relative humidity, which is assumed to be 1 where there is snow cover, and $P_{sfc}$ (Pa) is the surface pressure.

The RF induced by BC deposition is computed as follows:

$$RF = SW_{down}(SNOALB_0 - SNOALB_{bc}) \tag{6}$$

where $SNOALB_0$ is the snow albedo without any impurities, and $SNOALB_{bc}$ is the snow albedo that included BC.

In this study, the calculations of the surface energy balance were coupled into the Polar-WRF/Noah-MP.

## 2.4 BC and Snow Process

To evaluate the snow processes and land–atmosphere interactions related to BC in snow, we assume that the mixing ratio of BC is uniformly distributed throughout the snowpack. Additionally, a fixed mixing ratio of BC was established to eliminate the influence of varying BC concentrations on the SDE. Previous observations of BC in Arctic snow indicate that the concentration of BC can range from less than 5 to several hundred ng g$^{-1}$ in spring (Doherty et al., 2010; Zhang et al., 2024).

Furthermore, BC can accumulate on the snow surface during the melting season due to its insolubility (Forsström et al., 2013). As a result, the BC concentration in Arctic snow may exceed the reported values from the initial melting (Doherty et al., 2013; Dou et al., 2019). Taking all these factors into account, a BC mixing ratio of 50 ng g$^{-1}$ was chosen for Arctic snow. This value is appropriate for evaluating the potential effects of snow processes and land-atmosphere interactions on SDE, and it is a realistic estimate.

In this study, the mixing ratio of BC in the snowpack is fixed and remains unaffected by the processes of snowfall and snowmelt. Consequently, the changes in snow albedo induced by BC are directly influenced by snowpack properties, such as snow depth, snow density, and snow ice content. These properties can vary during the snowmelt process, thereby impacting the SDE (He & Ming, 2022). The evolution of snowpack properties is computed using the multi-layer snow module of Noah-MP(Niu et al., 2011). In Noah-MP, the snowpack can be divided into up to three layers, depending on the total snow depth

(See **Appendix A3**). The snow ice content and snow liquid water content in each snow layer are updated whenever melting or refreezing occurs, provided that the snowpack has explicit snow layers (i.e., snow depth $\geq$ 2.5 cm). If snow is present but its thickness is insufficient to form an explicit snow layer, the snow ice content and snow liquid water content will no longer be calculated, and all liquid water in the snow is assumed to be ponded on the soil surface (He et al., 2023). The snow ice content and snow liquid water content is connected to the processes of freezing and melting, which in turn influences snow density,

depth, and other properties, ultimately affecting the impact of the SDE.

    In the Noah-MP, the evolution of snowpack properties, including snow ice and liquid water content, snow thickness, and water flux out of snowpack bottom. If the snow layer temperature is higher than freezing point (273.15 K), then the snow layer ice is melting; if snow layer liquid water content is greater than 0, and snow layer temperature is lower than freezing point, then ice is refreezing. Once melting or freezing active, the snow ice amount will be updated. The amount of phase-change

water is computed as:

$$\Delta W_{phase}(i) = \frac{H_{M,phase}(i) \times \Delta t}{C_{LH,fus}} \tag{7}$$

where $\Delta W_{phase}$ (kg m$^{-2}$) is amount of phase-change water, $i$ is the snow layer, $H_{M,phase}$ (W m$^{-2}$) is the energy residual (surplus or loss), and it is computed as:

$$H_{M,phase}(i) = \frac{T_{snso}(i) - T_{frz}}{\Delta t} \times C_{h,snow} \times \Delta z \tag{8}$$

where $T_{sno}$ (K) is the snow temperature, $T_{frz} = 273.15$ (K) is the freezing point, $\Delta z$ (m) is the thickness of snow layer, $C_{h,snow}$ (J m$^{-3}$ K$^{-1}$) is the volumetric specific heat capacity of snow and it is calculated as:

$$C_{h,snow} = C_{h,ice} \times \theta_{ice,sno} + C_{h,wat} \times \theta_{liq,sno} \tag{9}$$

where $C_{h,ice}$ (J m$^{-3}$ K$^{-1}$) and $C_{h,wat}$ (J m$^{-3}$ K$^{-1}$) are the volumetric specific heat capacity of ice and water, respectively, $\theta_{ice,sno}$ and $\theta_{liq,sno}$ are partial volume of ice and liquid water in snow layer, respectively.

For each snow layer, if the freezing is active, then the snow ice content ($W_{ice,sno}$, [kg m$^{-2}$]) is updated as:

$$W_{ice,sno,new}(i) = \min\left(W_{snow,old}(i), W_{ice,sno,old}(i) - \Delta W_{phase}(i)\right) \tag{10}$$

If the melting is active, then the snow ice content ($W_{ice,sno}$, [kg m$^{-2}$]) is updated as:

$$W_{ice,sno,new}(i) = \max\left(0, W_{ice,sno,old}(i) - \Delta W_{phase}(i)\right) \tag{11}$$

Then, the snow liquid water content ($W_{liq,sno}$, [kg m$^{-2}$] is updated as:

$$W_{liq,sno,new}(i) = \max\left(0, W_{snow,old}(i) - W_{ice,sno,new}\right) \tag{12}$$


As the snow melts, the amount of liquid water content in the snowpack will rise, leading to an increase in snow density. Once the liquid water content surpasses the snowpack's maximum capacity to hold water, the snowpack will start to flow out, resulting in a reduction in snow depth. These changes in snow properties will influence the snow albedo reduction caused by BC.

**2.5. Model Configuration**

The Polar WRF domain (**Fig. 1**) is set to a polar stereographic grid centered at 90°N and had 220×220 grids. The spatial resolution is selected to 27km × 27 km, and there were 50 levels in the vertical from the surface to 10 hPa. The 27 km resolution is consistent with the ERA5 reanalysis data to ensure the accuracy of large-scale meteorological conditions, and it is significantly higher than the usual resolution employed in global climate models (which is typically over 1°) in earlier research 245 (e.g., Dou et al., 2012; Jiao et al., 2014; Ren et al., 2020). The initial meteorological fields in the model are derived from the fifth generation European Centre for Medium-Range Weather Forecasts (ECMWF) atmospheric reanalysis data (ERA5), available every 3 h, at 0.25° × 0.25° spatial resolution (https://www.ecmwf.int/en/forecasts/dataset/ecmwf-reanalysis-v5). The snow depth data are used the National Centers for Environmental Prediction (NCEP) operational Global Data Assimilation System (GDAS) final analysis data with a horizontal resolution of 0.1° × 0.1° for every 3 h 250 (https://rda.ucar.edu/datasets/ds084.4/). The sea ice thickness, snow depth on sea ice, and the sea ice albedo data are provided by National Snow and Ice Data Center (NSIDC, https://nsidc.org/data/data-access-tool/RDEFT4/). They are derived from the ESA CryoSat-2 Synthetic Aperture Interferometric Radar Altimeter (SIRAL), utilizing an enhanced waveform fitting algorithm (Kurtz et al., 2014). These data are presented on a 25 km grid as 30-day averages for the months of September through May. The physical parameterization options applied in this study are based on Hines and Bromwich (2017) and Hines 255 et al. (2019), including the new version of the rapid radiative transfer model for general circulation models (RRTMG) for both shortwave and longwave radiation (Iacono et al., 2008), the Morrison 2-moment scheme for cloud microphysics, Kain-Fritch convective scheme, and the polar-optimized Noah-MP land surface model (Bromwich et al., 2013; Niu et al., 2011). For the boundary layer, the Mellor-Yamada-Nakanishi-Niino (MYNN) level 2.5 PBL scheme with the MYNN surface layer scheme is used.

In addition to surface processes, atmospheric conditions like the boundary layer and clouds play a key role in effectively simulating precipitation and snowfall, which can influence the reliability of the simulation outcomes. As a result, choosing the appropriate boundary layer and cloud microphysics schemes is essential. In this study, the Mllor-Yamada-Nakanishi-Niino (MYNN) level 2.5 PBL scheme and the Morrison 2-moment cloud microphysics scheme were selected. Their performances in the Arctic are widely have been widely tested and verified (e.g., Hines & Bromwich, 2017; Hines et al., 2019; Turton et al., 265 2020; Xue et al., 2021).The MYNN model is a kind of second-order closure model that was proposed by Nakanishi and Niino (Nakanishi and Niino 2004, 2006, 2009) and is formulated as a modification of the Mellor-Yamada closure model (Mellor and Yamada 1982). In comparison to the MYNN level-3 scheme, the MYNN 2.5-level scheme retains the significant performance on the stable boundary layer simulations and reduces the computational cost (Kitamura, 2010; Nakanishi & Niino, 2009). The new version of the MYNN 2.5-level scheme implemented in WRF/PWRF Version 4.1.1 can improve downward shortwave 270 radiation at the surface (Olson et al., 2019), which is a key factor in assessing the reduction in snow albedo caused by BC deposition and the corresponding changes in the surface energy balance.

The Morrison 2-moment cloud microphysics scheme is a double-moment microphysics scheme that parameterizes the mixing ratio and number concentration of hydrometeors, covering cloud droplets, rain, ice crystals, snow, and graupel (Morrison & Gettelman, 2008). In the Polar-WRF, its droplet concentration is reduced from 250 cm−3 to 50 cm−3, which is

more applicable to polar regions (Hines & Bromwich, 2017). It has been extensively tested and has shown a great simulation capabilities, especially in the representation of mixed-phase clouds in the Arctic (Arteaga et al., 2024; Cho et al., 2020).

The simulation period spans from April 10 to May 15, 2020, coinciding with the Arctic snowmelt season. During this time, there is extensive snow cover and stronger solar radiation, leading to more pronounced impacts of the SDE by BC deposition. The first five days are considered the model spin-up time and are not analyzed. Some observations are used to

assess the model's simulation capabilities. The simulated meteorological parameters, including temperature, relative humidity, u and v wind speeds are compared with in-situ observation data released by the National Oceanic and Atmospheric Administration, (NOAA, https://gml.noaa.gov/dv/data/) at Barrow (156.6°W, 71.3°N;11 m a.s.l.) and Summit (38.5°W, 72.6°N; 3238 m a.s.l.). The sounding data used in this study are from the Department of Atmospheric Science, University of Wyoming (https://weather.uwyo.edu/upperair/sounding.html). The observed downward shortwave radiation, sensible heat flux

and latent heat flux in Alaska (149.3°W, 68.6°N) were downloaded from the Arctic Data Center (https://arcticdata.io/). This measurement was conducted by the University of Alaska Fairbanks (UAF) and is part of the Arctic Observing Network (AON) project. In this measurement, an eddy covariance system was employed to measure the fluxes of $CO_2$, water, and energy, and it was positioned on a 3-meter-high tripod in the center of the site. More details of the data are described by Bret-Harte et al. (2021).

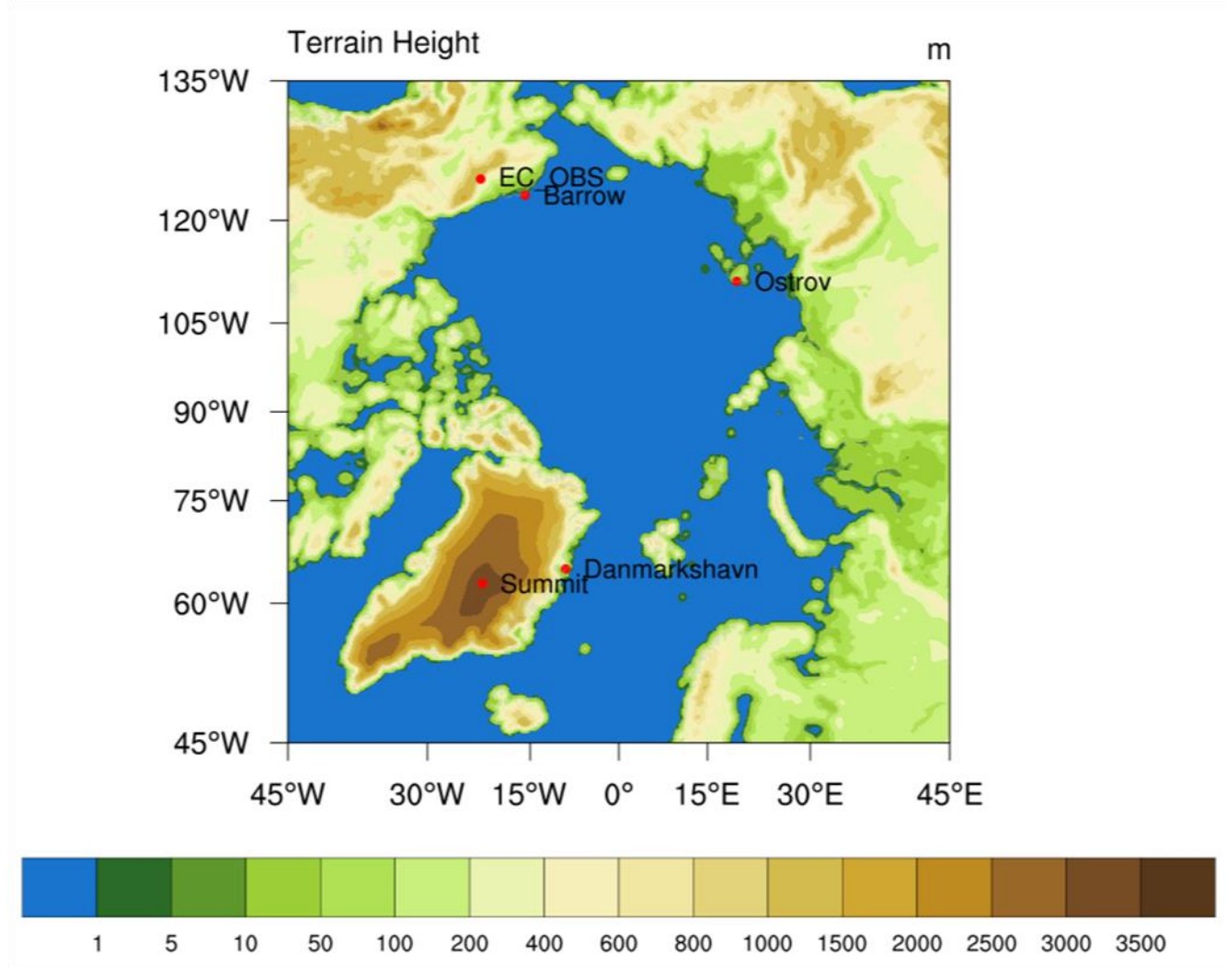


**Figure 1. Polar-WRF domain and terrain height. The red circles indicate the locations of the observation sites chosen for model evaluation in this study. The data from the Barrow and Summit stations are used to validate the performance of the modeled meteorological fields; the simulated vertical profiles of temperature, relative humidity, and wind speed are compared with the observed data from the Danmarkshavn and Ostrov stations; and EC_OBS is**

**the location of the surface energy observed by Bret-Harte et al. (2021) .**

**2.6. Experimental Design**

Two surface feedback processes are considered in this paper: the aging and melting processes of snow, in which changes in snow density and snow depth directly affect the reduction in snow albedo caused by BC, and the interaction between the land and atmosphere. The feedback processes between land and atmosphere can influence the impact of BC on the surface energy balance. In this study, the effects of these two key processes on snow albedo reduction caused by BC are assessed by comparing the differences in results between offline and online experiments. Several experiments have been designed (**Table 1**) to understand how BC deposition affects the surface energy exchange process in the Arctic and the importance of atmospheric processes and snow properties for the SDE. SNICAR-OFF is the offline coupling simulation, it was used to calculate the baseline of the SDE induced by BC deposition without snow cover change and land–atmosphere exchange process. The online simulation (SNICAR-ON) is fully coupled with Polar-WRF and the impacts of the SDE at every model timestep are computed by contrasting the dirty and clean snow albedo under the current surface and atmospheric conditions. Discrepancies between SNICAR-OFF and SNICAR-ON outcomes demonstrate the importance of surface feedback processes in comprehensively assessing the impacts of SDE. The SEN experiments are offline sensitivity experiments for the SNICAR model. SEN1-3 were designed to test the effects of snow density, snow depth, and snow grain size on the reduction of snow albedo caused by BC, respectively. SEN4 was designed to test the impact of the distribution of BC within snow on BC-induced changes in snow albedo. The purpose of the SEN experiments is to test the sensitivity of SNICAR to snow properties and to preliminarily assess the impact of snow characteristics on the reduction in snow albedo caused by BC.

**Table 1. Summary of model simulations**

| Name | Mixing ratio of BC (ng g$^{-1}$) | Snow density (kg m$^{-3}$) | Snow depth (m) | Snow grain radius (μm) | BC distribution with snow | Surface feedback processes |
|---|---|---|---|---|---|---|
| CTL | 0 | Provided by NCEP GDAS final analysis data | Provided by NCEP GDAS final analysis data | 100 μm (new snow) or 1000 μm (old snow) | Vertically uniform distribution | Included |
| SNICAR-OFF | 50 | Same as CTL | Same as CTL | Same as CTL | Same as CTL | Not included |
| SNICAR-ON | 50 | Same as CTL | Same as CTL | Same as CTL | Same as CTL | Included |
| SEN1 | 50 | Same as CTL | Same as CTL | Set as 50, 150, 250,500,100 | Same as CTL | Not included |
| SEN2 | 50 | Set as 100, 200,500 | Same as CTL | Same as CTL | Same as CTL | Not included |
| SEN3 | 50 | Same as CTL | Set as 0.05, 0.1,0.25,0.5,1.0 | Same as CTL | Same as CTL | Not included |
| SEN4 | 50 | Same as CTL | Same as CTL | Same as CTL | BC at top 5 cm layer | Not included |

## 3. Results

### 3.1. Evaluation of model performance

Meteorological parameters, such as the 2-m temperature and near-surface wind speed, are important factors of surface energy balance. In order to validate the performance of modeled meteorological fields at near-surface layers, the model results of CTL experiment were compared with in-situ observation data. The CTL experiment includes all surface feedback processes and is close to real-world simulation. The comparison results are shown in **Fig. 2** and the corresponding statistical metrics, including mean bias (MB), root mean square error (RMSE), and correlation coefficient (R), are listed in **Table S1**. The modeled 2-m temperature matches well with the observations at both stations (Rs > 0.9), except that some of the extremely high and low values appear abruptly. For relative humidity, the model captures the temporal variations fairly well at the Barrow station (Rs > 0.8), but a relatively high bias is found at the Summit station (Rs = 0.68, RMSE = 6.06). The mismatch in relative humidity between observations and simulations can be attributed to differences in the planetary boundary layer and surface parameterization settings in the model schemes (Qian et al.,2016). The simulated wind speed agrees well with the measurements, with Rs greater than 0.8, but the model underestimates the wind speed at both o stations (the MBs are -0.23 m s$^{-1}$ at the Barrow stations and -0.84 m s$^{-1}$ at the Summit station). For wind direction, the north-west wind at Barrow and southwest wind at Summit are generally captured by the simulation results. Similar bias of relative humidity and wind can also be found in other studies (e.g., Chen et al.,2021;Wilson et al.,2021).

The observed and simulated vertical profiles of temperature, specific humidity, and wind speed from the Danmarkshavn (18.7°W, 76.8°N; 12 m a.s.l.) and Ostrov (137.9°E, 76.0°N; 8 m a.s.l.) stations at 00:00 and 12:00 (UTC) averaged over the simulation periods are also shown in **Figs. S1** and **S2**. Generally, the model captures the vertical profiles of temperature quite well at both stations. However, the performances of the u-wind speed and v-wind are not as good as that of the temperature, and underestimation of the wind speeds are found for both the u and v components.

Surface feedback processes such as land–atmosphere exchange and changes of snowpack have already coupled into the Polar-WRF/Noah-MP and their performances have been widely validated (e.g., Justino et al., 2019; Li et al., 2022; Smith et al., 2017). To better access the impacts of these two surface feedback processes on the reduction in snow albedo caused by BC deposition and the corresponding changes in the surface energy balance, the modeled downward shortwave radiation, sensible heat flux (HS) and latent heat flux (LH) are also compared with the observation data in Alaska (149.3°W, 68.6°N), details of the data are described by Bret-Harte et al. (2021). The results are shown in **Fig. 3.,** and the statistical parameters of the observations and simulations are listed in **Table S2**. The simulation results agree well with the observed downward shortwave radiation; the MB is 0.24 W m$^{-2}$ and the R is 0.88. The evident diurnal variations in HS and LH are also reproduced. However, the overestimation of the HS and the underestimation of the LH were observed. These biases may result from the inaccurate land surface characteristics (e.g., land cover and topography) used in this study and the coarse model resolution. In summary, the Polar-WRF model reproduces the spatial-temporal evolutions of both meteorological fields and surface heat balance components fairly well, which provides confidence for further investigations.

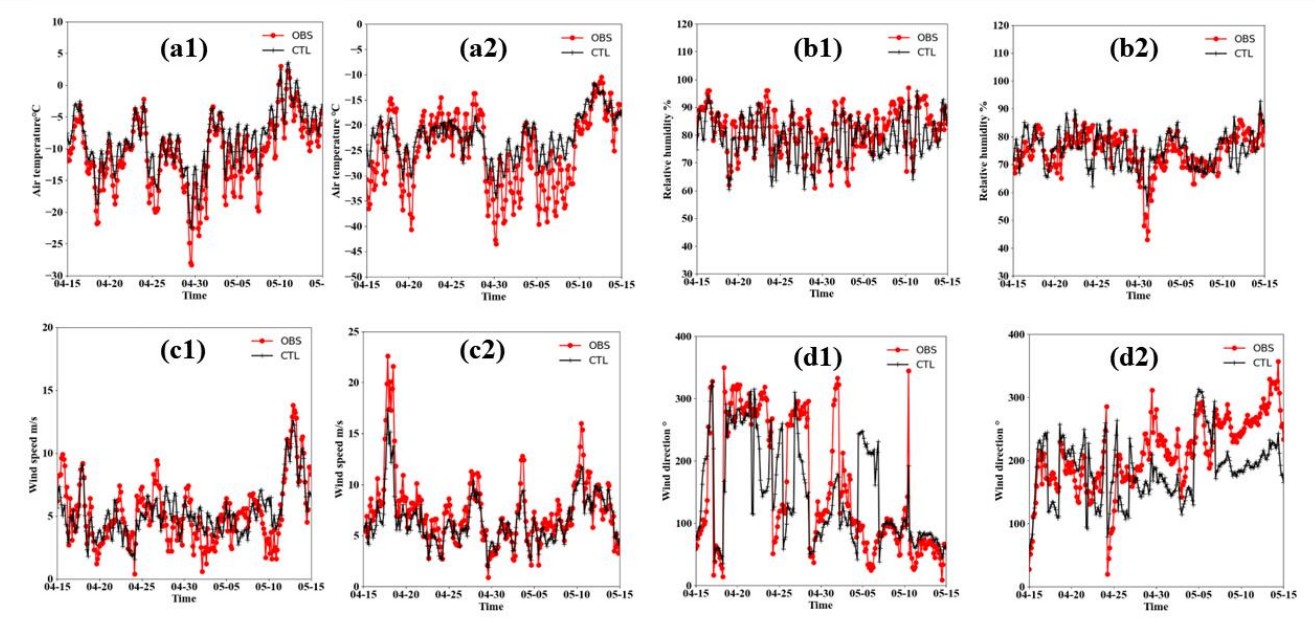

Figure 2. Time series of observed and simulated three-hourly average 2-m temperature (℃), 2-m relative humidity (%), 10-m wind speed (m s$^{-1}$), and 10-m wind direction (°) at the Barrow station (a1-d1) and Summit station (a2-d2) from 15 April to 15 May 2020.

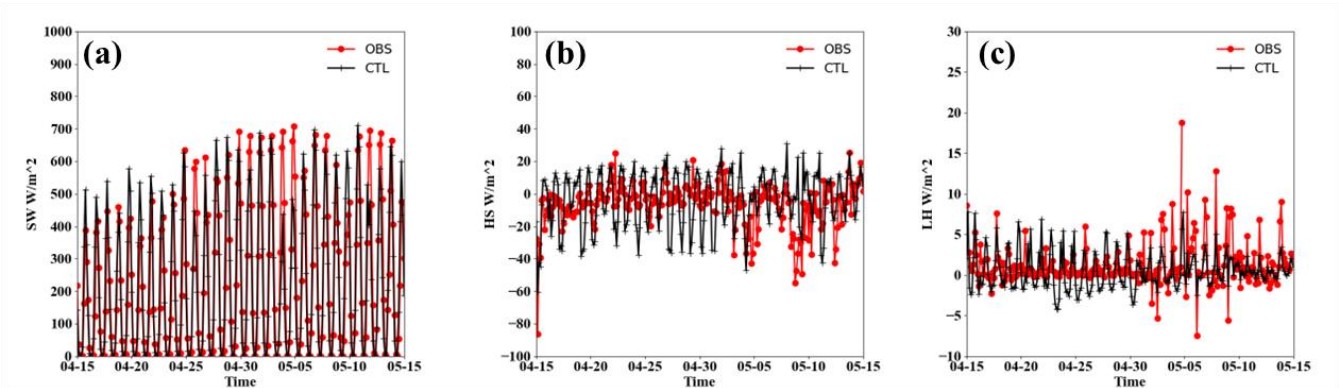

Figure 3. Time series of observed and simulated three-hourly average (a) incident solar radiation (SW) (W m$^{-2}$), (b) sensible heat flux (HS) (W m$^{-2}$) and (c) latent heat flux (LH) (W m$^{-2}$) in Alaska from 15 April to 15 May 2020. Positive (negative) values indicate gain (loss) by the atmosphere.

### 3.2. Sensitivity tests

Apart from the concentration of BC in snow, there are other factors that also influence snow albedo reduction induced by BC, including snow depth, snow density, snow effective grain radii and the vertical profile of BC in snow (Dang et al., 2015; Dang et al., 2017; Wang et al., 2014). **Fig. S3** shows the relative changes in albedo reduction for different snow depths, snow properties and vertical profiles of BC in snow from SEN experiments. The reduction in snow albedo caused by BC is affected by the size of snow grains in the visible band (Dang et al., 2016; Warren & Wiscombe, 1980). As shown in **Fig. S3a,** the magnitude of snow albedo reduction induced by BC deposition increases with increasing snow grain radius. As snow ages, the size of snow grains progressively increases over time (Colbeck, 1982), and this change in snow grain size has significant implications for the effects of BC deposition within snowpack..

Similarly, the manifestation of the SDE by BC is more pronounced for old snow characterized by high snow density (**Fig. S3b**). The density of snow tends to increase as it ages or undergoes various processes, such as melting and refreezing (Brun, 1989) , consequently contributing to a reduction in snow albedo. (Flanner et al., 2021; He & Ming, 2022). As shown in **Fig.S3**, compared with BC in a freshly fallen snowpack (snow density of 100 kg m$^{-3}$), BC in a snowpack with high density (500 kg m$^-$

[3]) can result in a maximum threefold greater reduction in snow albedo within the visible band. Snow depth has also emerged as a pivotal factor influencing the SDE induced by BC. Deeper snowpacks contain greater amounts of BC, resulting in more substantial impacts on albedo reduction (**Fig. S3c**). **Fig. S3d** shows the impacts of snow albedo reduction by BC using the different vertical profiles of BC in snow. Notably, applying the BC concentration of the surface layer (top 5 cm) to the entire snow column yields a snow albedo closely resembling that of the column-mean BC profile. These findings highlight the

importance of the BC concentration in surface layers for determining the SDE from BC deposition.

### 3.3 Spatial distribution of the SDE due to BC deposition

As discussed in Section 3.2, the direct albedo reduction caused by BC in snow is also influenced by factors other than the LAP concentration (Dang et al., 2015; Dang et al., 2017; He et al., 2018). Consequently, for a given BC concentration, the SDE resulting from BC and its impacts on the surface energy exchange process exhibit noticeable regional variations across

the Arctic (see **Figs. 4-5**). **Fig. 4** shows the mean snow albedo reduction induced by BC deposition. There are significant discrepancies between the results of the on-line and off-line simulations with the same mixing ratio of BC. In the case of SNICAR-OFF (the offline coupling simulation and the surface feedback processes are not included), a BC concentration of 50 ng $g^{-1}$ results in an average reduction of 0.0079 in broadband snow albedo across the Arctic. The most substantial impact of the SDE is observed in Greenland and Baffin Island, where the maximum decrease in snow albedo due to BC deposition

exceeds 0.01. Conversely, for SNICAR-ON (the online coupling simulation and the surface feedback processes are included), the SDE caused by BC is relatively weak, with the most pronounced change (>0.008) observed in Greenland and Eastern Siberia.

Fig. 5 illustrates the impacts of the SDE induced by BC deposition on the surface energy balance. The spatial distribution of the SDE-induced increase in daily-averaged RF closely corresponds to regions exhibiting significant reductions in snow

albedo for both SNICAR-OFF and SNICAR-ON (**Fig. 5a and Fig. 5e**). Changes in other heat balance components are also consistent with SDE-induced RF. For SNICAR-OFF (**Fig. 5b-d**), BC deposition at 50 ng $g^{-1}$ can cause an average radiative forcing of +1.6 W $m^{-2,}$ with a maximum of 4.7 W $m^{-2}$ in the Greenland region; upward longwave radiative changes are more pronounced in Greenland and the western Arctic of Russia, with a maximum of +0.55 W $m^{-2}$; changes in sensible heat transported by the surface to the atmosphere are stronger in the Greenland region, with a maximum of 3.3 W $m^{-2,}$ and changes

in latent heat occur in the Baffin Island region, with a maximum of 2.4 W $m^{-2}$. For SNICAR-ON (**Fig. 5f-h**), the simulated BC deposition with a concentration of 50 ng $g^{-1}$ can cause an average radiative forcing of +1.4 W $m^{-2,}$ with a maximum of 4.6 W $m^{-2}$ in Greenland and the western Russian Arctic; the upward longwave radiative changes are more pronounced in the western Russian Arctic, with a maximum of +0.51 W $m^{-2}$, while negative values of -0.54 W $m^{-2}$ occur in the eastern Russian Arctic. Sensible heat transport from the surface to the atmosphere is stronger in Greenland and the eastern Russian Arctic, with a

maximum of 3.7 W $m^{-2}$, and the maximum change in latent heat is observed in Greenland, reaching a maximum of 1.7 W $m^{-2}$.

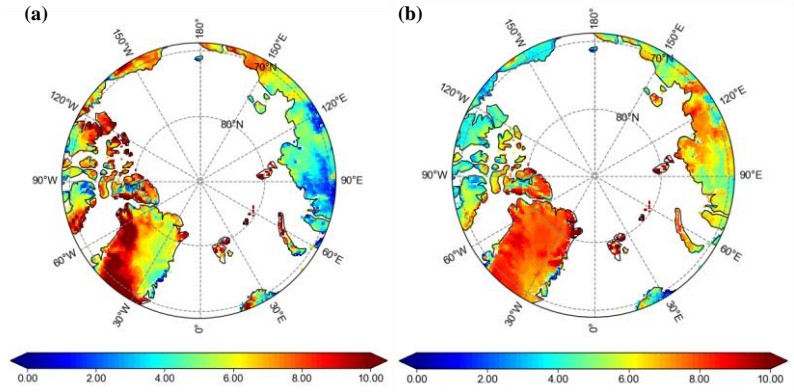

**Figure 4. Mean snow albedo reduction (10⁻³) induced by BC deposition in (a) SNICAR-OFF, and (b) SNICAR-ON.**

Similar spatial patterns are observed in both scenarios, with the most pronounced impacts of the SDE occurring in Greenland and relatively weaker impacts observed in the western Russian Arctic (**Fig. 6**). The spatial variability in the impacts of the SDE induced by BC is primarily attributable to regional disparities in snow conditions, as elucidated in Section 3.2. **Fig. 7** presents the distributions of snow depth and snow density across various Arctic regions, revealing that regions with greater snow depth, such as Greenland, exhibit more pronounced impacts of the SDE. Conversely, in the western Russian Arctic, where snow depths are shallow and snow densities are low, the impacts of SDE resulting from BC deposition are comparatively weaker. Additionally, within the SNICAR-ON simulations, substantial impacts of the SDE are observed in the eastern Russian Arctic, attributable to the greater snow density and relatively greater snow depth prevalent in the region.

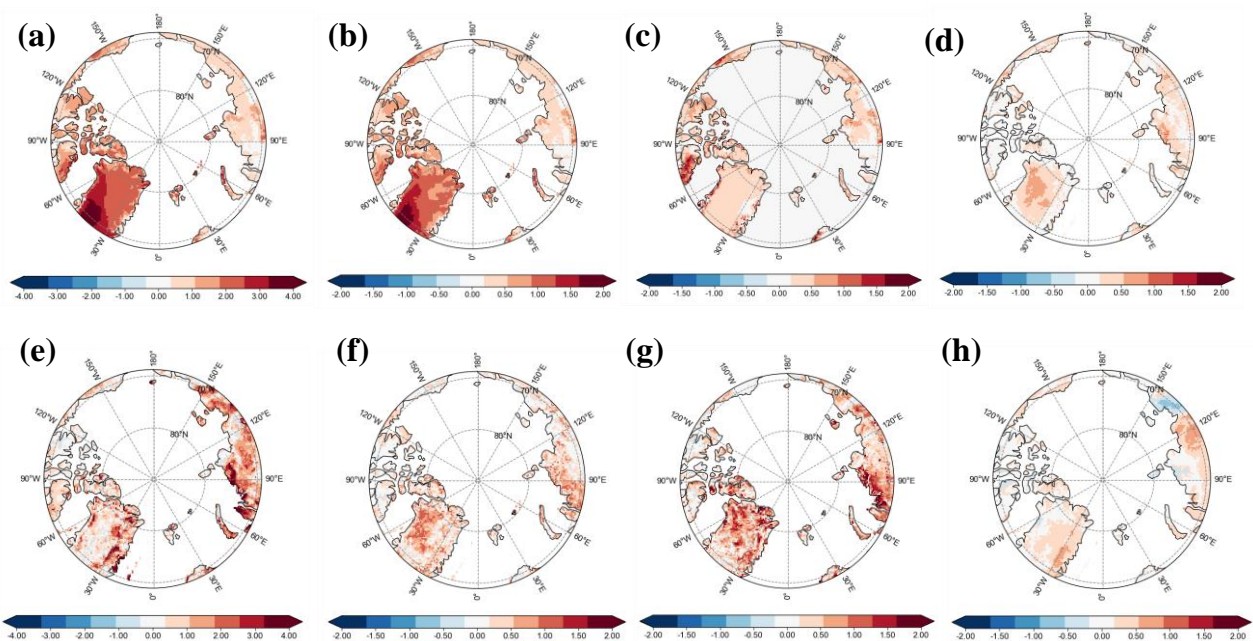

**Figure 5. Mean change in (a) RF (W m⁻²), (b) sensible heat flux (HS) (W m⁻²), (c) latent heat flux (LH) (W m⁻²), and (d) surface upwelling longwave radiation (LW) (W m⁻²) induced by BC deposition from SNICAR-OFFThe same as (e–h), respectively, except for SNICAR-ON. Positive (negative) values indicate gain (loss) by the atmosphere apart from RF.**

The snow albedo reduction induced by BC deposition is determined by a complex combination of multiple factors, including the properties of snowpack (e.g., snow depth and snow density), the underlying surface albedo and the incoming solar radiation (Dang et al., 2017; Flanner et al., 2021; Lin et al., 2024). In our simulation, we maintain a consistent mass mixing ratio of BC and uniform microphysical properties of snow particles. Thus, the mass of BC in snow is directly related to the snow depth, emerges as the primary determinant of the intensity of the SDE due to BC deposition. As discussed above, regions characterized by deeper snow, such as Greenland, demonstrate a more pronounced impact of SDE. In general, snow depth exhibits a nonlinear relationship with snow albedo (Flanner et al., 2021; Zhong et al., 2017). As snow depth increases, the total optical depth of the snowpack also increases. However, when the snowpack becomes sufficiently optically thick, photons are unable to pass through the medium without being absorbed or reflected. Consequently, beyond this threshold, further increases in snow depth no longer have an effect on the reduction in albedo caused by BC in snow. To better understand

the relationship between the SDE and snow depth, statistical analyses are conducted to examine the correlation between the snow albedo reduction induced by BC and snow depth across three distinct snow conditions (deep, moderate, and shallow), as delineated by Niu et al. (2011) (see **Appendix A3**).

Based on the results from SNICAR-OFF (the offline coupling simulation and the snow processes are not included), the relationship between the snow albedo reduction caused by vertically uniform distribution of 50 ng g$^{-1}$ BC and snow depth are

observed. **Fig. 8a** shows the statistical relationships between the snow albedo reduction and snow depth across various Arctic regions under shallow, moderate, and deep snow conditions. For shallow snow **(Fig. 8a1),** a distinct linear relationship is evident between the reduction in snow albedo and snow depth, characterized by an R-squared value exceeding 0.85 and a correlation coefficient below -0.9. Hence, a higher mass of BC in snow is closely correlated with a more pronounced impact of the SDE when the snow depth is shallow. For moderate snow (**Fig. 8a2**), a weak linear relationship is observed between the

reduction in snow albedo and snow depth, yielding a correlation coefficient of approximately 0.5 and an R-squared value of approximately 0.4. For deep snow condition (**Fig. 8a3**), the snow albedo reduction induced by BC and the impacts of SDE reach maximum levels.

Similar statistical relationships between the RF due to BC deposition and snow depth are also found, as shown in **Fig. 8b**. For shallow snow conditions (**Fig. 8b1**), a distinct linear relationship is evident between the RF and snow depth, characterized

by an R-squared value exceeding 0.7 and a correlation coefficient greater than 0.85. However, as snow depth increases (**Fig. 8b1-b2**), the linear relationships weaken, with significantly smaller correlation coefficients (<0.4). These findings suggest that snow depth plays a key role in evaluating the mass of BC on the SDE. Particularly during melting periods, the SDE caused by BC deposition may vary as the snow melts, warranting further investigation.

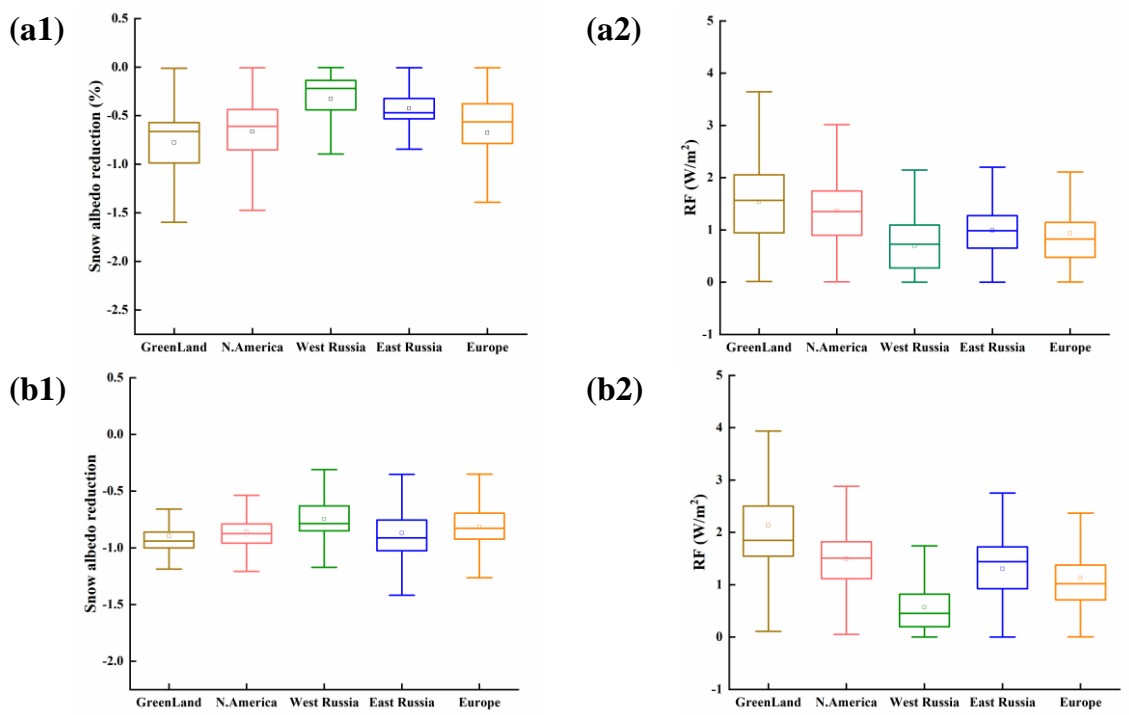


**Figure 6. Snow albedo reduction and RF induced by BC in snow from SNICAR-OFF (a1-a2) and SNICAR-ON (b1-b2) across different Arctic regions.**

### 3.4 Temporal Evolution of the SDE caused by BC Deposition

As discussed above, the impacts of the SDE can be affected by changes in snowpack properties through two primary mechanisms. First, this study assumed that the distribution of BC was uniform throughout the snowpack, indicating that deeper layers of snow contained higher concentrations of BC particles. The overall mass of BC present in snow can influence the modeling effects of the SDE. In addition, as snow ages and begins to melt, the effects of the SDE induced by BC increase due to the reduced snow albedo (He & Ming, 2022). To analyze the impact of the evolution of snow on the SDE, this study

categorized snow into three types on the basis of snow depth and investigated the related changes on the SDE as snow ages and melts. The three snow conditions are determined on the basis of the total snow depth ($h_{sno}$): shallow ($0.025m < h_{sno} < 0.25m$), moderate ($0.25m \leq h_{sno} < 0.45m$) and deep ($h_{sno} \geq 0.45m$). The detailed information is shown in **Appendix A3**. Based on SNICAR-ON simulation results (include the snow processes), the temporal evolution of the SDE caused by a fixed 50 ng g$^{-1}$ BC has been studied. **Fig. 9** illustrates the temporal evolution of changes in the surface energy balance induced by

BC deposition. Apparent diurnal variations can be found in all changes in the surface heat balance components. As shown in **Eq.6**, the RF resulting from BC deposition is strongly dependent on the incoming solar radiation. In general, solar radiation exhibits distinct diurnal and seasonal variations, which can significantly influence the RF attributed to BC deposition. Consequently, these variations can affect the impacts of the SDE on the surface energy exchange process in the Arctic. **Fig. 9a** shows the temporal evolution of the RF induced by BC. Noticeable diurnal variations in RF are evident and the most

pronounced influence of the SDE occurs at the peak sun elevation (**Fig. 10a**).

The temporal trends of the impacts of the SDE due to BC deposition under different snow conditions are also shown in **Fig. 9** and **Fig. 10b**. At the beginning of the simulation, the impacts of the SDE are relatively weak due to the lower solar radiation (**Fig. 10a**) and the freshness of the snowpack. As the snow ages and incident solar radiation increases, the impacts of SDE caused by BC have also increase. However, as the snow melts and snow depths decrease, a gradual decrease in snow

albedo caused by BC is observed for moderate and shallow snowpacks, resulting in a weakened SDE. Conversely, in the case of a deep snowpack, the snow depth remains sufficiently deep throughout the melting period. A decrease in snow depth has little impacts on the albedo reduction. Moreover, BC-induced changes in snow albedo are augmented as snow ages **(Fig. 10b)**.

The increased absorption of incident solar radiation by BC deposition alters the surface heat balance components through various mechanisms. **Fig. 11a and Fig. 11b** depict the changes in surface temperature and snow melt induced by BC deposition

respectively. These changes can directly influence the surface energy balance. For the HS (**Fig. 9b**), a similar diurnal variation pattern to that of the RF is observed. As illustrated in **Eq.2,** the HS is governed primarily by the temperature difference between the surface and the atmosphere. With decreasing in snow albedo, the surface absorbs more solar radiation leading to an increase in surface temperature (**Fig. 11a**), and consequently enhancing the HS towards the atmosphere. The LH is influenced by both temperature and humidity differences between the surface and atmosphere (**Eq.3**). Thus, the change in the LH induced by the

SDE is influenced not only by the increase in surface temperature but also by the process of snow melting. The diurnal variation in the LH **(Fig. 9c)** also coincides with RF, exhibiting relatively low values. However, as the snow melts faster (**Fig. 11b**) and surface runoff increases, the change in the LH may gradually increase over a longer time scale. (Lau et al., 2018). The LW is directly related to the increase in surface temperature. However, due to the high specific heat capacity of snow, the increase in surface temperature caused by BC deposition on snow cover is minimal. Therefore, the LW change is negligible (**Fig. 9d**).


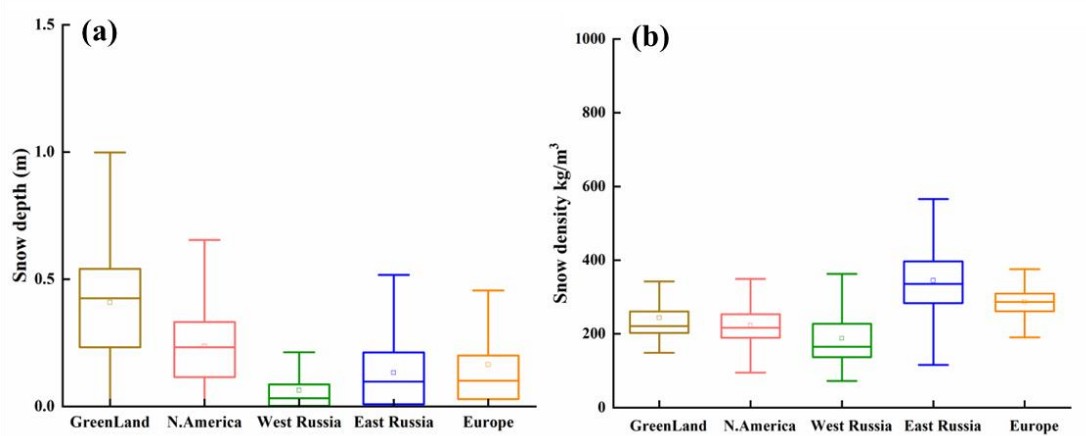

**Figure 7. Distribution of (a) snow depth (m) and (b) snow density (kg m$^{-3}$) across different Arctic regions.**

In addition to the surface variables, the impacts of SDE by BC deposition can also influence the near-surface air through land-atmosphere interactions. **Fig. 11c** and **Fig. 11d** illustrate the changes in 2-m temperature and 2-m specific humidity induced by BC deposition respectively. At the onset of the simulation, the 2-m temperature exhibits minimal change until approximately a week into the simulation. As the energy exchange between the surface and the near-surface air progresses, the air temperature gradually increases. A similar pattern is observed for the 2-m specific humidity. Near-surface humidity can be directly influenced by snowmelt. Initially, snowmelt remains unchanged (**Fig. 11b**). However, as the snow albedo decreases and more solar radiation penetrates the surface, the internal energy and liquid content of the snowpack increase. When the liquid water fraction of the snowpack exceeds the maximum allowable snowpack liquid mass fraction, water fluxes out of the snowpack, accelerating the rate of snowmelt (He et al., 2023), and leading to an increase in the near-surface specific humidity.

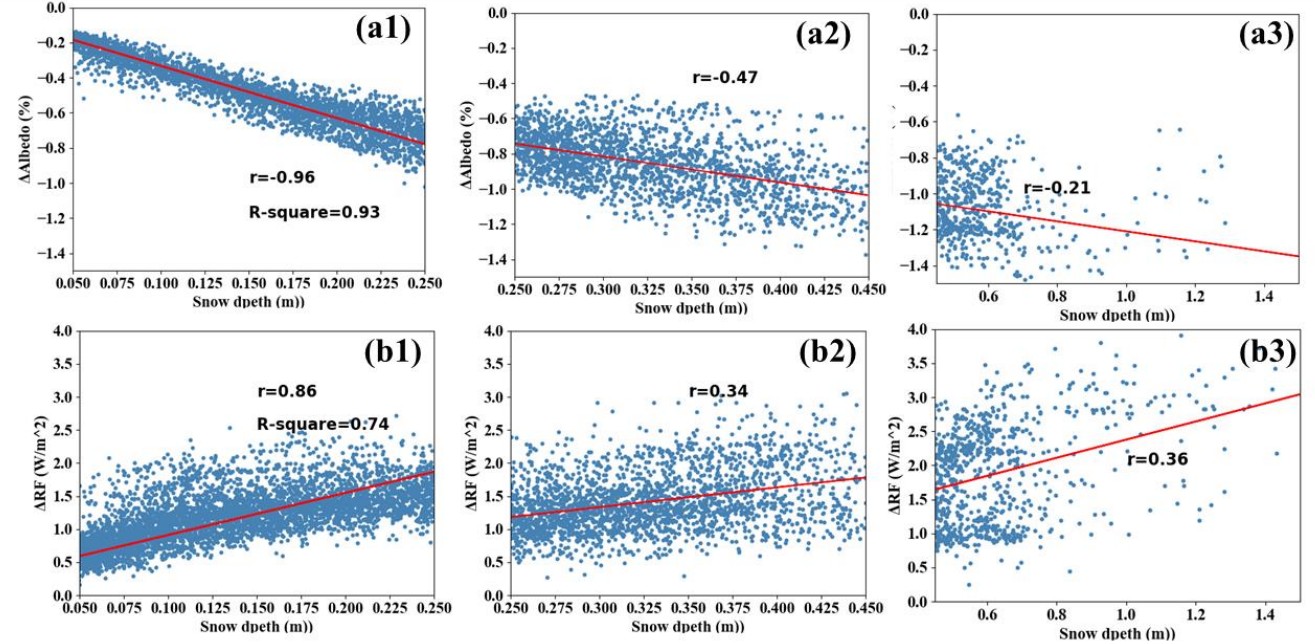

**Figure 8. Relationship between (a1-a3) the snow albedo reduction (%) induced by BC deposition and snow depth (m) and between (b1-b3) the RF (W m$^{-2}$) and snow depth (m)**

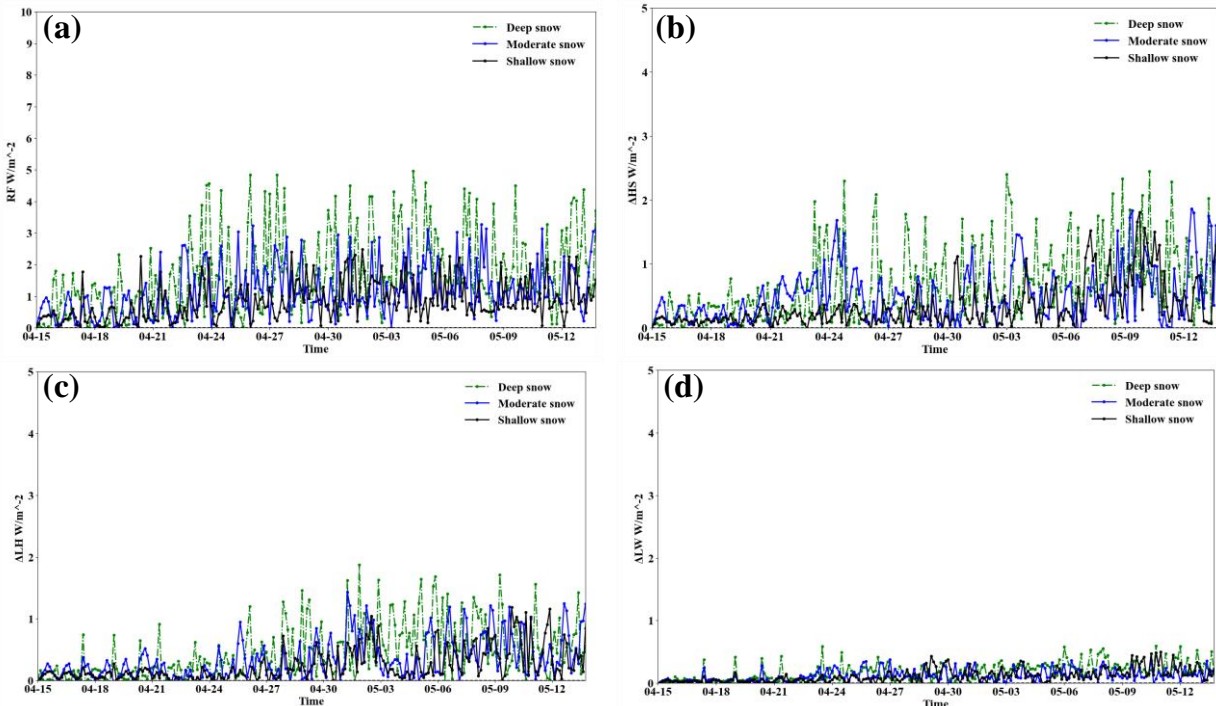

**Figure 9.** Averaged three-hour on-line simulation results of the change in the (a) RF (W m$^{-2}$) (b) sensible heat flux (HS) (W m$^{-2}$) (c) latent heat flux (LH) (W m$^{-2}$) (d) surface upwelling longwave radiation (LW) (W m$^{-2}$) induced by BC deposition in different snow layers. Positive (negative) values indicate gain (loss) by the atmosphere apart from RF.

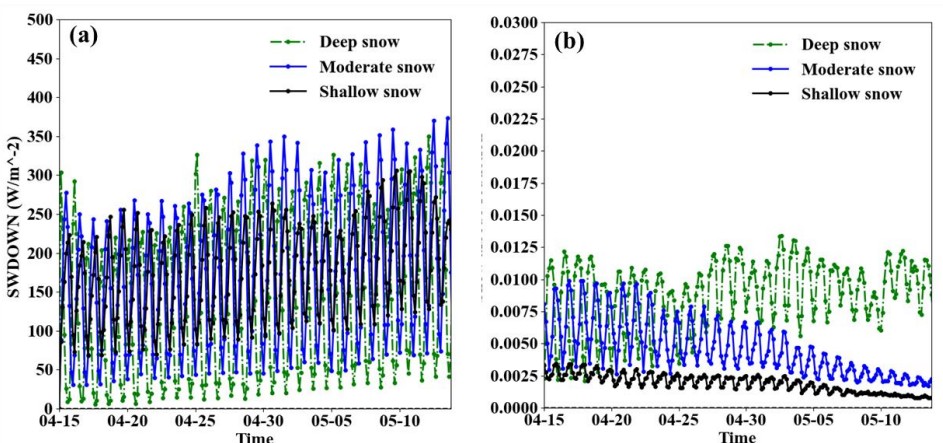

**Figure 10.** Averaged three-hour on-line simulation results of the (a) incident solar radiation (W m$^{-2}$) and (b) snow albedo reduction induced by BC deposition in different snow layers.

As discussed above, the SDE resulting from BC is closely associated with snow depth. The average changes in surface energy under various snow conditions are listed in **Table 2**. Notably, regions with deep snow experience more significant impacts from the SDE due to BC deposition, despite not necessarily receiving the highest levels of incident solar radiation (**Fig. 10a**). This can be explained by the duration of the SDE, as depicted in **Fig. 10b**. The snow albedo reduction induced by BC deposition remains relatively high throughout the simulation in deep snow regions. Conversely, in shallow and moderate snow regions, the snow albedo reduction resulting from BC gradually decreases as snow melts until it is completely melted. For a given column-mean BC concentration in snow, the impacts of SDE are approximately 25-41% greater in deep snow-covered areas than in shallow snow-covered areas, leading to a 19-40% increase in snowmelt. These findings underscore the

importance of addressing BC deposition in regions characterized by deep snow (e.g., Greenland), as deep snow may have a more pronounced impact on surface energy exchange.

**Table 2. Average changes in heat balance components (W m$^{-2}$); surface skin temperature, TSK (K); 2m air temperature, T2 (K); and snow melt (mm d$^{-1}$) in different snow layer regions. SW, LW, LH, and HS represent incoming solar**
**radiation, longwave radiation, latent heat flux, and sensible heat flux, respectively. Positive (negative)values indicate gain (loss) by atmosphere. The H$_m$ is computed as the residue of all heat balance components.**

|  | Deep snow | Moderate snow | Shallow snow |
|---|---|---|---|
| Snow albedo | -0.0089 | -0.0057 | -0.0021 |
| RF (W m$^{-2}$) | -1.64 | 1.21 | 0.97 |
| LW (W m$^{-2}$) | 0.15 | 0.16 | 0.11 |
| HS (W m$^{-2}$) | 0.61 | 0.43 | 0.38 |
| LH (W m$^{-2}$) | 0.46 | 0.30 | 0.26 |
| H$_m$ (W m$^{-2}$) | 0.42 | 0.32 | 0.22 |
| TSK (K) | 0.078 | 0.061 | 0.047 |
| T2 (K) | 0.064 | 0.046 | 0.031 |
| Snow melt (mm d$^{-1}$) | 0.072 | 0.058 | 0.033 |

### 3.5 Differences in physical mechanisms between off-line and on-line methods

The impacts of the BC-induced SDE on the Arctic surface and near-surface air from both the off-line and on-line
experiments are summarized in **Table 3**. The impacts of SDE by BC are greater for SNICAR-OFF than for SNICAR-ON. The disparities between the off-line and on-line simulations can be attributed to their distinct physical mechanisms. As described in **Table 1**, the off-line experiment (SNICAR-OFF) does not incorporate surface and atmospheric processes related to the impacts of the SDE on the surface energy balance. In contrast, all relevant processes are included in the on-line experiment (SNICAR-ON), making its mechanisms more comprehensive and closer to real-world conditions. Consequently, the disparities
observed in their outcomes serve to elucidate the significance of these associated processes.

**Table 3. Average changes in heat balance components (W m$^{-2}$); surface skin temperature, TSK (K); and snow melt (mm d$^{-1}$) during the simulation period. SW, LW, LH, and HS represent incoming solar radiation, longwave radiation, latent heat flux, and sensible heat flux, respectively. Positive (negative)values indicate gain (loss) by atmosphere. The**
**H$_m$ is computed as the residue of all heat balance components.**

|  | SNICAR-ON | SNICAR-OFF |
|---|---|---|
| Snow albedo | -0.0068 | -0.0079 |
| RF (W m$^{-2}$) | -1.4 | -1.6 |
| LW (W m$^{-2}$) | 0.14 | 0.16 |
| HS (W m$^{-2}$) | 0.48 | 0.55 |
| LH (W m$^{-2}$) | 0.34 | 0.38 |
| H$_m$ (W m$^{-2}$) | -0.44 | -0.51 |
| TSK (K) | 0.067 | 0.071 |
| Snow melt (mm d$^{-1}$) | 0.069 | 0.078 |

As previously emphasized, snowpack conditions, especially snow depth play a key role in BC-induced albedo reduction. In Sections 3.2 and 3.3, an evident correlation between the impacts of SDE by BC and snow depth was established. **Fig. 12** provides a typical example comparing the disparity in snow albedo reduction due to BC between the off-line and on-line
simulations. According to the SNICAR-ON simulations (the snow processes are included), the impacts of SDE by BC vary

with snowfall and snowmelt processes. For instance, at the beginning (April 15-20), as the snow depth decreases due to snowmelt, the impact of the SDE due to BC deposition weakens. However, with subsequent snowfall (April 21-April 23), the impacts of the SDE are enhanced due to the increase in snow depth. In contrast, the modeled snow albedo reduction by BC of the SNICAR-OFF(the snow processes are not included) is not influenced by changes in snow conditions. Therefore, the

reduction of snow albedo remains constant in response to incident solar radiation. As a result, the decrease in modeled snow albedo caused by BC in SNICAR-OFF is approximately 16.2% greater than that by SNICAR-ON on average, leading to stronger modeled impacts of the SDE.

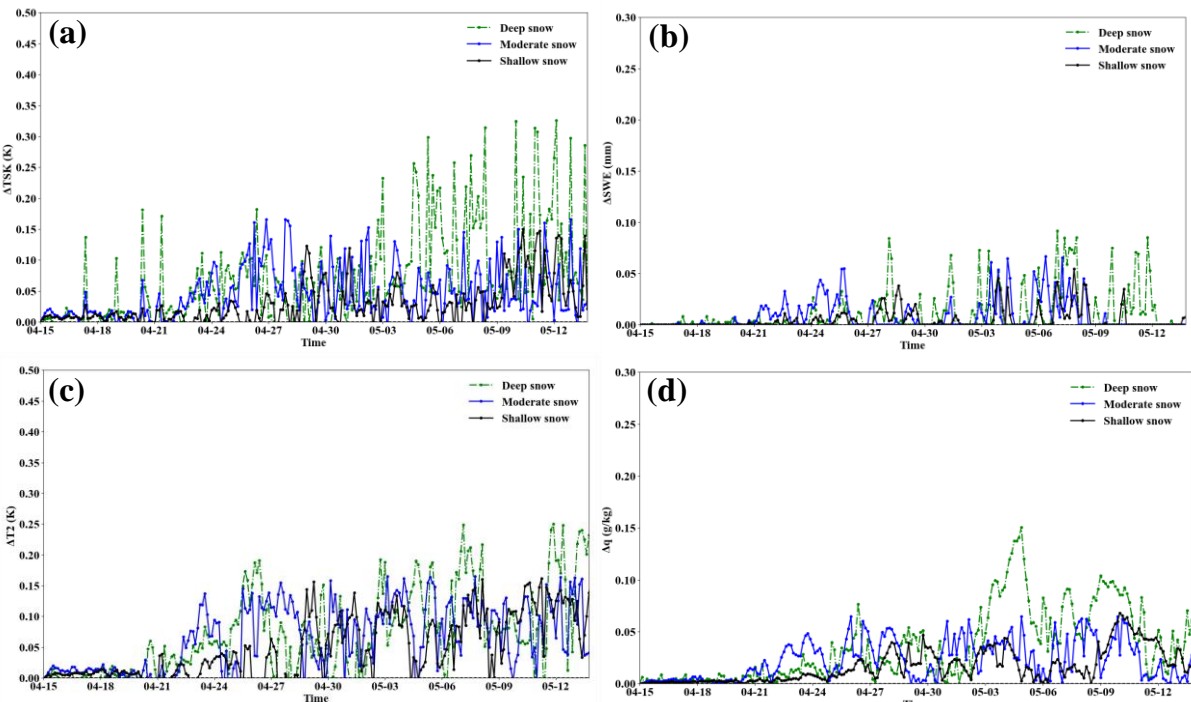

**Figure 11. Averaged three-hour on-line simulation results of the changes (a) surface temperature (K) (b) snow melt (mm h⁻¹) (c) 2-m air temperature (K) and (d) 2-m specific humidity (g kg⁻¹) induced by BC deposition in different snow layers.**

Land-atmosphere interactions are another crucial process affecting the modeled impacts of the SDE induced by BC. The

changes in surface energy balance induced by BC are not only influenced by surface variables but also controlled by near-surface air conditions. **Fig. 13** illustrates the impacts of land-atmosphere interactions on the modeled changes in surface energy balance. As defined in **Eq.2**, sensible heat, the transfer of heat from the surface to the atmosphere without any phase change, is dependent on the temperature difference between the surface and near-surface air. Fig. **13a** shows the temperature difference between the surface and the near-surface air from SNICAR-ON (in black line, the land-atmosphere interactions are included)

and SNICAR-OFF (in green line, the land-atmosphere interactions are not included). Initially, there is no apparent difference between SNICAR-ON and SNICAR-OFF. However, as the energy exchange process between the surface and the near-surface air progresses, the air temperature also increases (**Fig. 11c**), thereby reducing the transfer of sensible heat from the surface to the atmosphere due to the smaller temperature differences between the surface and the air. Hence, the modeled changes in the HS from SNICAR-OFF are 12.3-57.7% greater than those from SNICAR-ON during the simulation period.


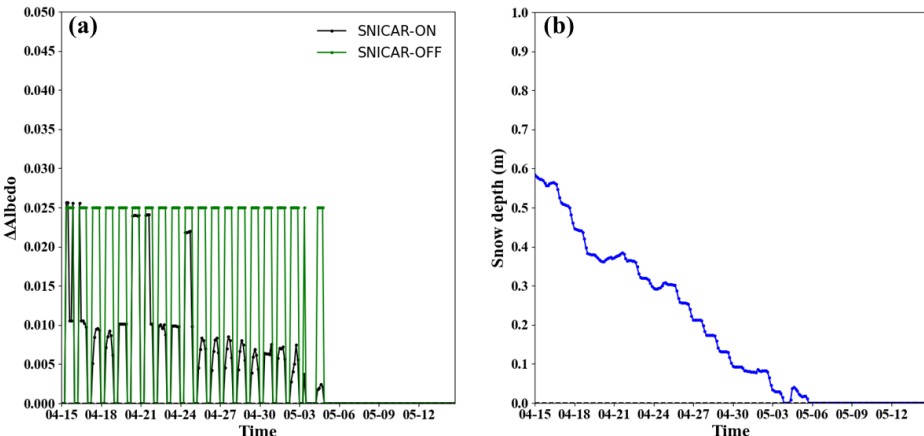

**Figure 12. (a) Averaged three-hour simulation results of the snow albedo reduction by BC deposition and (b) snow depth change during the period.**

A similar phenomenon has also been observed in the modeled LH changes. According to **Eq.3**, the surface latent heat flux is controlled by the specific humidity differences between the surface and the near-surface air. Excluding the effect of changes in temperature discussed above, the specific humidity in the near-surface air is closely related to snowmelt. **Fig. 13b** depicts specific humidity differences between the surface and near-surface air from SNICAR-ON (black line) and SNICAR-OFF (green line). When BC is deposited on the snow surface and reduces its albedo, the snow absorbs more solar radiation, leading to accelerated snowmelt (Kang et al., 2020), An increase in snowmelt (**Fig. 11d**) can increase the near-surface specific humidity, consequently resulting in lower changes in latent heat flux. Therefore, the modeled LH changes from SNICAR-OFF are 8.7-51.4% greater than those from SNICAR-ON during the simulation period.

In summary, the differences in the modeled impacts of the SDE by the same BC concentrations in snow between the on-line and off-line simulations can be explained by two main processes: snowmelt and land-atmosphere interactions. **Fig. 14** shows how the two processes affect the impacts of SDE due to BC deposition. As discussed above, the off-line simulation tends to overestimate the impacts of the SDE, by up to more than 50% sometimes due to the lack of relevant processes. Therefore, it is crucial to consider all relevant atmospheric and surface processes to accurately estimate the impacts of SDE by BC, particularly its effects on the surface energy balance.

## 4. Conclusions

By comparing off-line and on-line coupled simulations between Polar-WRF and SNICAR, this study investigated the critical mechanisms and key factors influencing changes in surface heat transfer considering the impacts of the SDE induced by BC deposition in the Arctic. First, the performances of the modeled meteorological fields and surface energy balance were validated by comparing them with in-situ and ground-based observation data. The simulation results generally captured the values and variation trends of the observation data. To test the sensitivity of the SNICAR model and explore the factors influencing the impacts of the SDE at given BC concentrations, several sensitivity tests were conducted. The results indicated that snowpack properties, such as snow depth, snow density, and the snow grains size, can affect snow albedo reduction caused by BC. Moreover, similar spatial distribution characteristics of impacts of the SDE induced by BC from off-line and on-line coupling simulations were found, with more pronounced impacts observed in regions with greater snow depth and density, such as Greenland and Eastern Siberia. In SNICAR-OFF, a clear relationship between snow depth and snow albedo reduction by BC was also observed and discussed. Additionally, the temporal evolutions of SDE impacts on both surface and near-surface air as snow melts and ages was investigated by SNICAR-ON. Finally, based on the above findings, two main physical mechanisms affecting the impacts of the SDE on surface energy balances were highlighted by comparing the results from

online and offline coupled simulations, aiming to provide valuable suggestions for accurately assessing the impacts of the SDE by deposition in the Arctic. The four main conclusions are summarized as follows:

1.  The simulation results indicate that BC deposition can directly affect the surface energy balance by decreasing snow albedo and its corresponding daily-averaged RF. On average, BC deposition at 50 ng g$^{-1}$ can cause RF values of 1.6 W m$^{-2}$ and 1.4 W m$^{-2}$ according to the SNICAR-OFF and SNICAR-ON configuration, respectively. Similar spatial patterns are observed in both simulations, with the most pronounced impacts of the SDE occurring in Greenland and relatively weaker impacts observed in the western Russian Arctic. The RF resulting from BC deposition can reach greater 4 W m$^{-2}$ and is primarily found in Greenland, Baffin Island, and East Siberia, regions characterized by deep snow depths and high snow densities.

2.  The impacts of the SDE due to BC are strongly influenced by snow depth. In SNICAR-OFF, when the snow depth is shallow, a clear linear relationship with a correlation coefficient exceeding 0.9 and an R-squared value greater than 0.85 between the snow depth and the reduction in snow albedo has been observed. As snow depth increases, the snow albedo reduction induced by BC and the impacts of the SDE gradually increase until reaching maximum values when the snowpack becomes sufficiently optically thick.

3.  Apparent diurnal variations can be found in all changes in the surface heat balance components: in SNICAR-ON, the impacts of the SDE increases as the incident solar radiation increases, and the most pronounced influences occur at the peak sun elevation. The impacts of the SDE tend to increase as snow ages and decrease as snow melts. Regions with deep snowpack, such as Greenland, tend to exhibit greater sensitivity to BC deposition due to the higher mass of BC in snow and the longer duration of the SDE. For a given column-mean BC concentration in snow, the impacts of the SDE are approximately 25-41% greater in deep snow-covered areas than in shallow snow-covered areas, leading to a 19-40% increase in snowmelt in SNICAR-ON.

4.  Compared to the offline coupling simulation, snowmelt and land-atmosphere interactions have significant impacts on assessing changes in the surface energy balance caused by BC deposition. The impacts of the SDE due to BC deposition diminish gradually as snow melts. On average, the decrease in snow depth due to snow melt can offset 16.2% of decrease in snow albedo caused by BC, and this decrease can reach more than 50% during periods of accelerated snowmelt. Near-surface air temperature and specific humidity can also be influenced by BC deposition through land-atmosphere interactions, and changes in near-surface air meteorological factors can reduce the HS by 12.3-57.7% changes in HS and the LH by 8.7-51.4%.

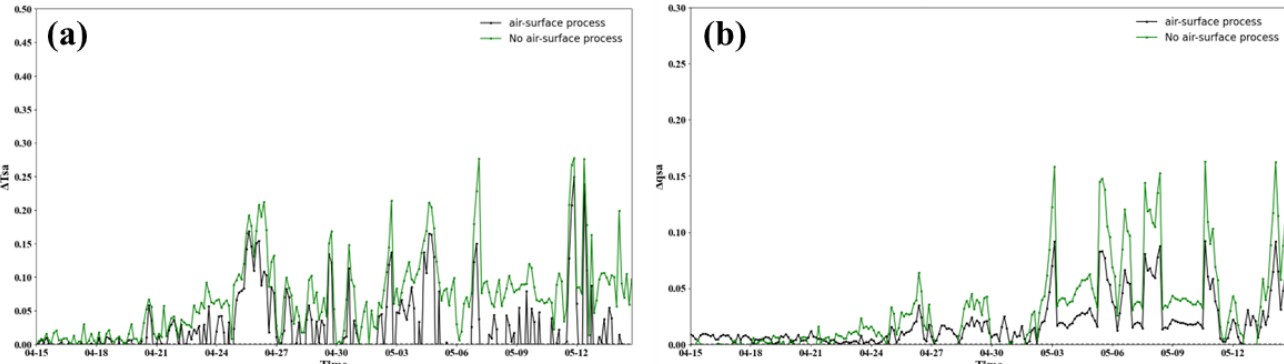

**Figure 13. Differences between the (a) temperature (K) and (b) specific humidity (g kg$^{-1}$) of the surface and air during the air-surface exchange process (black line) and without the air-surface process (green line)**

There are several uncertainties and limitations in this study. For instance, the evolution of snow effective grain size is not considered, although it may significantly influence snow reduction (Dang et al., 2015; Flanner et al., 2021). Additionally, BC

can accumulate on snow surfaces during melt amplification due to its insolubility (Doherty et al., 2010; Forsström et al., 2013). The accumulation of BC in snow can also affect the SDE. Moreover, earlier snow melting resulting from the SDE induced by BC deposition may alter atmospheric circulation and the cloud fraction, leading to significant changes in Arctic climate (Jiang et al., 2016; Lau et al., 2018). Clouds also play an important role in assessing the SDE, such as their radiative effects and impact on snowfall. However, studying clouds in the Arctic presents several challenges (AMAP, 2021; Huang et al., 2010). This research primarily emphasizes surface feedback processes while overlooking the role of clouds. These areas warrant further research to better understand their implications for Arctic climate dynamics.

Overall, this study emphasizes the importance of considering all relevant atmospheric and surface processes, especially the processes of snow melting and land-atmosphere interactions, to accurately estimate the impacts of the SDE on surface energy exchange. In addition, understanding the temporal evolution of the SDE is also crucial for comprehending how BC deposition affects surface energy exchange in the Arctic. Previous studies have predominantly estimated the impacts of the SDE on the Arctic by calculating the average RF due to BC deposition (Chen et al., 2022; Dang et al., 2017; Dou et al., 2012), which may not fully capture the impacts of the SDE from BC deposition on Arctic climate change. Therefore, future research should prioritize high-resolution modeling studies that incorporate detailed physical processes to enhance our understanding of the impacts of the SDE on Arctic climate change.

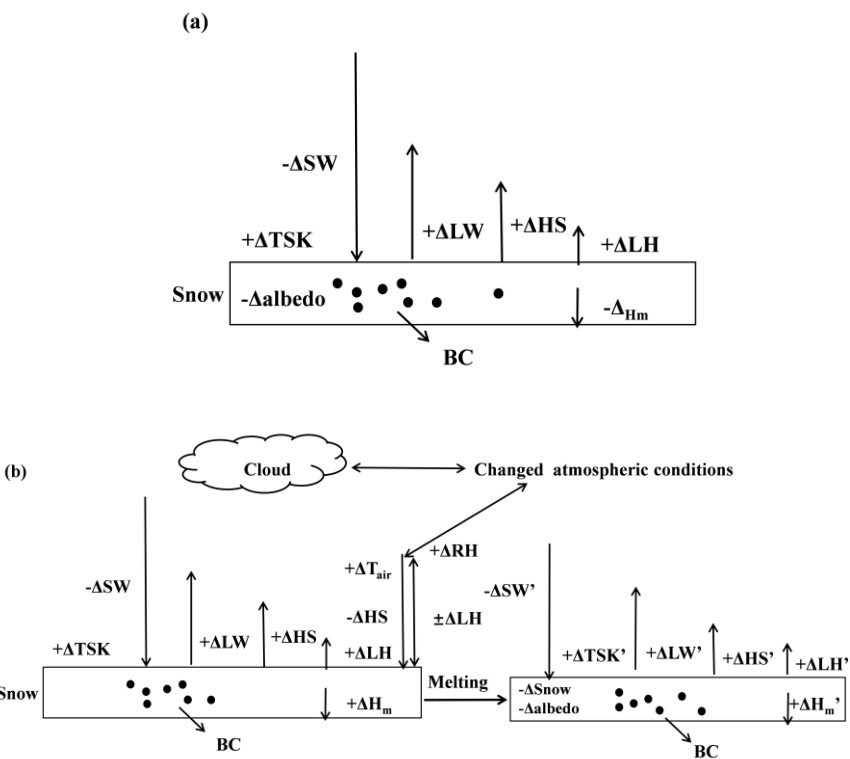

**Figure 14. Conceptual diagram of SDE due to BC deposition on the surface energy exchange process: (a) off-line (b) on-line**

**Appendix**

**A1. CLASS scheme**

In CLASS, the snow albedo is calculated as follows:

$$SNOALB = SNOALB_{old}^t + \frac{\min(Q_{snow}dt, swe_{mx}) \times (0.84 - SNOALB_{old}^t)}{swe_{mx}} \tag{A1}$$

where $SNOALB$ is the snow albedo, $SNOALB_{old}^t$ is the snow albedo before snowfall, $swe_{mx} = 1$ mm is the critical value of the new snow water equivalent, which is assumed to fully cover the old snow the $Q_{snow}$ is the snowfall rate (mm s$^{-1}$), and $dt$ is the time step.

The snow albedo is assumed to be no less than 0.55. The process of determining the now age is expressed by an exponential function of the modeling time step:

$$SNOALB_{old}^t = 0.55 + SNOALB_{old}^{t-1} - 0.55) \times \exp\left(\frac{-0.01dt}{3600}\right) \tag{A2}$$

where $SNOALB_{old}^{t-1}$ is the snow albedo before snowfall at the last time step.

## A2. BATS scheme

In BATS, the snow albedos of diffuse and direct radiation are different. For fresh snow, the snow albedo is 0.95 for the visible band and 0.65 for near-infrared band. They are calculated as follows:

$$SNOALB_{si1} = 0.95(1 - 0.2A_c) \tag{A3}$$
$$SNOALB_{si2} = 0.65(1 - 0.5A_c) \tag{A4}$$

where $SNOALB_{si1}$ and $SNOALB_{si2}$ are the snow albedos of diffuse radiation for the visible band and the near-infrared band respectively. $A_c$ is a factor of snow aging.

For direct radiation:

$$SNOALB_{sd1} = SNOALB_{si1} + 0.4Z_c(1 - SNOALB_{si1}) \tag{A5}$$
$$SNOALB_{sd2} = SNOALB_{si2} + 0.4Z_c(1 - SNOALB_{si2}) \tag{A6}$$

where $SNOALB_{sd1}$ and $SNOALB_{sd2}$ are the snow albedos of direct radiation for the visible band and for the near-infrared band, respectively. $Z_c$ is a factor of the solar zenith angle.

$Z_c$ is defined as follows:

$$Z_c = \frac{1.5}{1 + cosZ} - 0.5 \tag{A7}$$

Where Z is solar zenith angle.

The process of snow age determination is described as follows:

$$A_c = \frac{\tau_s}{1 + \tau_s} \tag{A8}$$

$$\tau_s^t = \tau_s^{t-1}\left\{1 - \frac{\max(0, \Delta swe)}{swe_{mx}}\right\} \tag{A9}$$

$$\Delta\tau_s = (\tau_1 + \tau_2 + \tau_3)10^{-6}dt \tag{A10}$$

$$\begin{cases} arg = 5000\left(\frac{1}{TFRZ} - \frac{1}{TG}\right) \\ \tau_1 = \exp(arg) \\ \tau_2 = \min(1, \exp(10arg)) \\ \tau_3 = 0.3 \end{cases} \tag{A11}$$

where TFRZ is the freezing temperature set to 273.16 Kin Noah-MP, TG is the ground temperature (K), and $\Delta swe$ is the difference in snow water equivalents between the current time step and the previous time step.

## A3. Snow Layers in Noah-MP

Based on the total snow depth ($h_{sno}$), the snowpack can be divided into as many as three layers. The detailed descriptions are shown in Yang and Niu (2003). When $h_{sno}$ is less than 0.045 m, there is no snow layer. When $h_{sno} \geq 0.025$ m, and less than 0.05 m, only one snow layer is created, and its thickness ($\Delta z_0$) is equal to $h_{sno}$. When $h_{sno} \geq 0.05$ m, two snow layers

are created and their thicknesses are equal. When $h_{sno} \geq 0.01$ m, the two-layer thicknesses are: 0.05 m and $h_{sno}$-0.05 m respectively. When $h_{sno} > 0.15$ m, a third layer is created; the three-layer thicknesses are: $\Delta z_{-2} = 0.05$ m, and $\Delta z_{-1} = \Delta z_0 = \frac{h_{sno}-0.05}{2}$ m; and when $h_{sno} \geq 0.45$ m, the layer thicknesses for the three snow layers are: $\Delta z_{-2} = 0.05$ m, $\Delta z_{-1} = 0.2$ m, and $\Delta z_0 = h_{sno}$-0.25 m.

Based on the snow layers in Noah-MP and the snowpack depths, three snow conditions are determined: shallow ($0.025m < h_{sno} < 0.25m$), moderate ($0.25m \leq h_{sno} < 0.45m$) and deep ($h_{sno} \geq 0.45m$). The three snow conditions are used to analyze the relationships between the total mass of BC in snow and its impact on SDE in Section 3.3 and 3.4.

## A4. List of abbreviations

| Abbreviation | Definition |
| --- | --- |
| BC | Black Carbon |
| SDE | Snow darkening effect |
| Polar-WRF | Polar-optimized version of the Weather Research and Forecasting model |
| SNICAR | Snow, Ice, Aerosol, and Radiation (SNICAR) model, in this study, the SNICAR was coupled in Polar-WRF/Noah-MP as a snow albedo parameterization to investigate the SDE by BC. |
| RF | Radiative Forcing |
| LSM | Land surfacing model |
| Noah LSM | A land surface model that simulates the interactions between the land and atmosphere. It has already been coupled with WRF as a land surface scheme. In Polar-WRF, the Noah-MP has been optimized for polar regions. |
| Noah-MP | The community Noah land surface model with multiple parameterization options. It is based on the Noah LSM developed by Niu et al. (3011). It has already been coupled with WRF as a land surface scheme. In Polar-WRF, the Noah-MP has been optimized for polar regions. |
| CLASS | The Canadian Land Surface Scheme snow albedo parameterization in Noah-MP |
| BATS | The Biosphere–Atmosphere Transfer Scheme snow albedo parameterization in Noah-MP |

*Author contribution,* ZZ and MZ initiated the study and designed the experiments. ZZ performed the simulations and carried out the data analysis. LZ and MZ provided useful comments on the paper. ZZ prepared the manuscript with contributions from all co-authors.

*Code availability.* The code of Polar-WRF can get from the https://polarmet.osu.edu/PWRF/registration.php, the SNICAR source can download in https://github.com/mflanner/SNICARv3, Polar-WRF/Noah-MP coupled with SNICAR code is available at https://github.com/ZhangZiLu0831/PWRF_NoahMP_SNICAR.

*Competing interests.* The authors declare that they have no conflict of interest.

*Financial support.* This paper is supported by the National Key Research and Development Program (Grant nos. 2022YFC2807203,2022YFB2302701).

*Acknowledgements.* The simulations conducted in this study were supported by the National Key Scientific and Technological Infrastructure project "Earth System Numerical Simulation Facility" (Earth Lab). We also thank Cenlin He (NACR) and Jinming Feng (CAS, IAP) for their help with code.

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
