# Peer review of "A numerical sensitivity study on the snow darkening effect by black carbon deposition over the Arctic in spring"

_EGUsphere, 2024_

## Author Comment (AC1)

*Reviewer 1:*

*General comments:*

*Black carbon (BC) aerosols have significant impacts on earth system radiative balance due to its strong light-absorbing properties in the visible wavelength. Once BC deposited in snow, it accelerates snow melting, and further impacts the climate. In this study, a physically based snow radiative model was coupled into Polar-WRF to investigate the SDE of BC in the arctic region. The topic is interesting, however, this manuscript cannot be published in the current state, the authors need to resolved several major issues before it can be reconsidered. Please see my comments in detail below.*

We thank the reviewer for the valuable comments and suggestions, which were very helpful for improving our manuscript. We revised the manuscript carefully, as described in our point-to-point responses to the comments.

*Comment 1:*

*Title of the manuscript underlines the time period is melting season, however I do not see any significance of snow melting in the manuscript. The biggest problem is that the authors did not take BC amplification effects during melting period into modeling. The authors have to solve this issue before it can be reconsidered by ACP. See Flanner et al., JGR, 112, D11202, doi:10.1029/2006JD008003, 2007 and Doherty et al., JGR, 118, 5553–5569, doi:10.1002/jgrd.50235, 2013.*

We sincerely appreciate the valuable comments. In this study, our primary aim is to assess the impacts of snowpack properties and land–atmosphere exchange on the reduction in surface snow albedo caused by black carbon (BC) deposition and the corresponding changes in the surface energy balance. Thus, we assumed a uniform distribution of BC in snow and designed a series of sensitivity experiments to evaluate the effects of the snowmelt process and land–atmosphere feedback on the BC-induced snow darkening effect (SDE), with a particular focus on the impacts on snow albedo and the surface energy balance. These experiments are based on an idealized assumption that the BC mixing ratio in snow remains constant and is uniformly distributed within the snowpack. In previous studies (Zhang et al., 2024), we collected observational data on BC in Arctic snow. In the future, we will evaluate BC amplification effects in the Arctic on the basis of observations and another numerical sensitivity study. However, in this study, we focused only on assessing the effects

of snowpack changes and land–atmosphere interaction processes on the SDE induced by BC, particularly the impacts on snow albedo and the surface energy balance.

References

Zhang, Z., Zhou, L., & Zhang, M. A progress review of black carbon deposition on Arctic snow and ice and its impact on climate change. Advances in Polar Science, 35(2), 178-191.·doi:10.12429/j.advps.2023.0024,2024

*Comment 2:*

*An assumption of 50 ppb was applied in this study, which largely deviates from the real world. The BC mixing ratios in the arctic can vary from <10 to several hundred ppb. See Doherty et al., ACP, 10, 11647–11680, 2010 and Doherty et al., JGR, 120, 11,391–11,400, doi:10.1002/2015JD024018, 2015. Therefore, I think the SDE values reported by the authors are not trusted.*

Thank you for your valuable comment. We sincerely appreciate the valuable comments. In this study, our primary aim is to assess the impacts of snowpack properties and land–atmosphere exchange on the reduction in surface snow albedo caused by black carbon (BC) deposition and the corresponding changes in the surface energy balance. There are significant regional differences in the impacts of BC deposition on snow albedo and the surface energy balance in the Arctic (Dou & Xiao, 2016; Kang et al., 2020). These differences are related not only to the varying amounts of BC deposition across regions but also to the spatial heterogeneity of surface characteristics such as snow distribution in the Arctic. In this study, we assumed an ideal scenario in which BC is uniformly distributed in snow at a fixed concentration. Numerical sensitivity experiments were then used to evaluate the significance of the snowmelt and land–atmosphere exchange processes. In the future, we will assess the changes in snow albedo caused by Arctic BC deposition on the basis of actual observational data (collected by Zhang et al.,2024), but here, we conducted only a numerical sensitivity study.

[Figure]

Figure 1. Locations and concentrations of BC snow observations collected from Arctic campaigns between 2005 and 2018 in spring (a) and summer (b). The figure is from Zhang et al. (2024).

References

Dou, T.-F., & Xiao, C.-D. An overview of black carbon deposition and its radiative forcing over the Arctic. Advances in Climate Change Research, 7(3), 115-

122. · doi:10.1016/j.accre.2016.10.003,2016

Kang, S., Zhang, Y., Qian, Y., & Wang, H. A review of black carbon in snow and ice and its impact on the cryosphere. Earth-Science Reviews,

210. · doi:10.1016/j.earscirev.2020.103346,2020

Zhang, Z., Zhou, L., & Zhang, M. A progress review of black carbon deposition on Arctic snow and ice and its impact on climate change. Advances in Polar Science, 35(2), 178-

191. · doi:10.12429/j.advps.2023.0024,2024

**Specific comments**

*Comment 1:*

*Abstract: Title of the manuscript is "Modeling study of the snow darkening effect by black carbon deposition over the Arctic during the melting period." However, I did not see any discussion related to the effects of melting on SDE due to BC in the abstract.*

Thank you for the valuable comments. In this study, our primary aim is to assess the impacts of snowpack properties and land–atmosphere exchange on the reduction in surface snow albedo caused by black carbon (BC) deposition and the corresponding changes in the surface energy balance. Other processes and impacts related to the SDE due to BC will be discussed in future

studies. In addition, to better reflect the purpose of the paper, the title was changed to "A numerical sensitivity study on the snow darkening effect by black carbon deposition over the Arctic in spring".

*Comment 2:*

*The abstract is unnecessarily long, needs to be shortened substantially.*

Thank you for the valuable comments. We reorganized our abstract as follows:

"The rapid warming of the Arctic, driven by glacial and sea ice melt, poses significant challenges to Earth's climate, ecosystems, and economy. Recent evidence indicates that the snow-darkening effect (SDE), caused by black carbon (BC) deposition, plays a crucial role in accelerated warming. However, high-resolution simulations assessing the impacts from the properties of snowpack and land–atmosphere interactions on the changes in the surface energy balance of the Arctic caused by BC remain scarce. This study integrates the Snow, Ice, Aerosol, and Radiation (SNICAR) model with a polar-optimized version of the Weather Research and Forecasting model (Polar-WRF) to evaluate the impacts of snow melting and land–atmosphere interaction processes on the SDE due to BC deposition. The simulation results indicate that BC deposition can directly affect the surface energy balance by decreasing snow albedo and its corresponding radiative forcing (RF). On average, BC deposition at 50 ng $g^{-1}$ causes a daily average RF of 1.6 W $m^{-2}$ in offline simulations (without surface feedbacks) and 1.4 W $m^{-2}$ in online simulations (with surface feedbacks). The reduction in snow albedo induced by BC is strongly dependent on snow depth, with a significant linear relationship observed when snow depth is shallow. In regions with deep snowpack, such as Greenland, BC deposition leads to a 25–41% greater SDE impact and a 19–40% increase in snowmelt than in areas with shallow snow. Snowmelt and land–atmosphere interactions play significant roles in assessing changes in the surface energy balance caused by BC deposition based on a comparison of results from offline and online coupled simulations via Polar-WRF/Noah-MP and SNICAR. Offline simulations tend to overestimate SDE impacts by more than 50% because crucial surface feedback processes are excluded. This study underscores the importance of incorporating detailed physical processes in high-resolution models to improve our understanding of the role of the SDE in Arctic climate change."

*Comment 3:*

*L21: instantaneous RF or daily-averaged RF here? Please clarify.*

Thank you for the comments. The RF in this study is calculated as the daily average over the simulation period on the basis of the model results. We also indicated this in the manuscript for better understanding.

*Comment 4:*

*L31: What is Noah-MP?*

Thank you for your comments. Land surface models (LSMs) are useful modeling tools for resolving terrestrial responses to and interactions with the atmosphere, ocean, glaciers, and sea ice in the Earth system. Traditionally, LSMs are thought to provide lower boundary conditions to coupled atmospheric models. The Noah-MP is a widely used land surface model and has been used in many studies and operational weather/climate models (HRLDAS, WRF, WRF-Hydro/NWM, NOAA/UFS, NASA/LIS, etc.). In this study, Noah-MP was coupled with Polar-WRF to investigate the impact of BC deposition on snow albedo and the surface energy balance.

*Comment 5:*

*I think the authors need to clarify in the introduction that why did they want to investigate the SDE during melting season instead of snow accumulation season or stable season? And proper citations are required.*

Thank you for your comments. In this study, we selected the spring snowmelt period as our simulation period for two main reasons. First, during the spring melt period, the relatively strong solar radiation combined with the near-maximum snowpack depth makes the impact of BC on the SDE particularly significant for the terrestrial Arctic surface (Doherty et al., 2010; Zhang et al., 2024). Additionally, during this period, the aging and melting processes of snow affect the reduction in snow albedo induced by BC (Dang et al., 2015; He & Ming, 2022), and the influence of snow processes on BC-induced changes in snow albedo has not been well evaluated. We also included the relevant content in the introduction of the manuscript (L80-85).

References

Dang, C., Brandt, R. E., & Warren, S. G. Parameterizations for narrowband and broadband albedo of pure snow and snow containing mineral dust and black carbon. Journal of Geophysical Research: Atmospheres, 120(11), 5446-5468. • doi:10.1002/2014JD022646,2015

Doherty, S. J., Warren, S. G., Grenfell, T. C., Clarke, A. D., & Brandt, R. E. Light-absorbing impurities in Arctic snow. Atmospheric Chemistry and Physics, 10(23), 11647-11680. • doi:10.5194/acp-10-11647-2010,2010

He, C., & Ming, J. Modelling light-absorbing particle–snow–radiation interactions and impacts on snow albedo: fundamentals, recent advances and future directions. Environmental Chemistry, 19(5), 296-311. • doi:10.1071/en22013,2022

Zhang, Z., Zhou, L., & Zhang, M. A progress review of black carbon deposition on Arctic snow and ice and its impact on climate change. Advances in Polar Science, 35(2), 178-191. • doi:10.12429/j.advps.2023.0024,2024

*Comment 6:*
*L98: SSNICAR->SNICAR*

We apologize for our carelessness, and we replaced it with "SNICAR".

*Comment 7:*
*L99: Please briefly describe surface feedbacks in the modeling experiments.*

Thank you for the comments. Two surface feedback processes are considered in this paper: the aging and melting processes of snow, in which changes in snow density and snow depth directly affect the reduction in snow albedo caused by BC, and the interaction between the land and atmosphere. The feedback processes between land and atmosphere can influence the impact of BC on the surface energy balance. In this study, the effects of these two key processes on snow albedo reduction caused by BC are assessed by comparing the differences in results between offline and online experiments. We also included the relevant content in the introduction of the manuscript (L304-309).

*Comment 8:*
*L105: Please delete "and" after "processes through".*

We apologize for our carelessness, and we deleted it.

*Comment 9:*

*L109: What is Noah-LSM? Same for CLASS and BATS in L113-114. I highly suggested the authors generate a list of Abbreviations in the appendix for the readers, too many model names in the manuscript.*

Thank you for the comments. Land surface models (LSMs) are useful modeling tools for resolving terrestrial responses to and interactions with the atmosphere, ocean, glaciers, and sea ice in the Earth system. Traditionally, LSMs are thought to provide lower boundary conditions to coupled atmospheric models. The Noah-LSM is a widely used land surface model and has been used in many studies and operational weather/climate models, such as WRF and the Polar-WRF.

CLASS is the Canadian Land Surface Scheme snow albedo parameterization in Noah-MP, and BATS is the Biosphere–Atmosphere Transfer Scheme snow albedo parameterization in Noah-MP. Both of them are snow albedo schemes originally used in Noah-MP, but none of them account for the impact of BC on snow albedo. Therefore, in this study, we coupled the SNICAR model as the snow albedo scheme with the Polar-WRF/Noah-MP to assess the effect of BC on snow albedo.

We apologize for our carelessness, and we generated a list of abbreviations in Appendix A4 as follows:

| Abbreviation | Definition |
| --- | --- |
| BC | Black Carbon |
| SDE | Snow darkening effect |
| Polar-WRF | Polar-optimized version of the Weather Research and Forecasting model |
| SNICAR | Snow, Ice, Aerosol, and Radiation (SNICAR) model, in this study, the SNICAR was coupled in Polar-WRF/Noah-MP as a snow albedo parameterization to investigate the SDE by BC. |
| RF | Radiative Forcing |
| LSM | Land surfacing model |
| Noah LSM | A land surface model that simulates the interactions between the land and atmosphere. It has already been coupled with WRF as a land |

| | |
|---|---|
| | surface scheme. In Polar-WRF, the Noah has been optimized for polar regions. |
| Noah-MP | The community Noah land surface model with multiple parameterization options. It is based on the Noah LSM developed by Niu et al. (3011). It has already been coupled with WRF as a land surface scheme. In Polar-WRF, the Noah-MP has been optimized for polar regions. |
| CLASS | The Canadian Land Surface Scheme snow albedo parameterization in Noah-MP |
| BATS | The Biosphere–Atmosphere Transfer Scheme snow albedo parameterization in Noah-MP |

References

Niu, G.-Y., Yang, Z.-L., Mitchell, K. E., Chen, F., Ek, M. B., Barlage, M., Kumar, A., et al. The community Noah land surface model with multiparameterization options (Noah-MP): 1. Model description and evaluation with local-scale measurements. Journal of Geophysical Research: Atmospheres, 116(D12).·doi:10.1029/2010JD015139,2011

*Comment 10:*

*Please think about remove Sec. 3.2. Similar results have been reported in other studies such as He, C. 2022, https://doi.org/10.1071/EN22013, and nothing interesting of the results.*

Thank you for the valuable comment. Although the content of this section has been previously reported, it is still necessary for the discussions in the following two sections. Therefore, we condensed this section and moved Figure 4 to the supplementary materials.

*Comment 11:*

*3.3: I do not agree with the authors that they used a fixed BC mixing ratios in Arctic snow, and the values about BC RF they reported are with very low confidence.*

Thank you for the valuable comment. In this numeric sensitivity study, our primary aim is to assess the impact of snowpack properties and land–atmosphere exchange on the reduction in

surface snow albedo caused by BC deposition and the corresponding changes in the surface energy balance. The assessment of the RF induced by BC is not the primary objective of this study, and it will be evaluated in future studies.

*Comment 12:*

*3.4: Where is the BC deposition time series data from? I did not see the data source.*

Thank you for the comments. In this numeric sensitivity study, we assumed an ideal scenario in which BC is uniformly distributed in snow at a fixed concentration. The BC deposition time series data were not used in this study.

---

## Author Comment (AC2)

*Reviewer 2:*

*General comments:*

***T****his study investigates the effect of black carbon (BC) on snow surfaces over the Arctic using the Snow, Ice, Aerosol, and Radiation (SNICAR) model online and offline with a polar-optimized version of the Weather Research and Forecasting model (Polar-WRF). The research focuses on an important problem as a better understanding of the impacts of snow conditions and land-atmosphere interactions is critical in more accurately assessing the overall impact of BC in the Arctic. However, there are a few major flaws in the manuscript.*

We thank the reviewer for the valuable comments and suggestions, which were very helpful for improving our manuscript. We revised the manuscript carefully, as described in our point-to-point responses to the comments.

*Major comments:*

*Comment 1:*

*The modeling seems detached from the reality. Only validation of meteorological factors is included. However, other variables, such as BC concentration, snow depth, surface albedo, runoff, etc, also need to be compared with observations to ensure the numbers are in line with those in the real Earth system in order to generate meaningful results. For example, it is unclear how 50 ng/g is selected as BC mixing ratio. For the existing validation, please provide explanations of the simulated biases, especially temperature and wind direction. Also, observations from the two locations cannot sufficiently represent the whole Arctic condition.*

Thank you for your valuable comments and advice. In this study, our primary aim is to assess the impacts of snowpack properties and land–atmosphere exchange on the reduction in surface snow albedo caused by black carbon (BC) deposition and the corresponding changes in the surface energy balance. Thus, we assumed a uniform distribution of BC in snow and designed a series of sensitivity experiments to explore the effects of the snowmelt process and land–atmosphere feedback on the BC-induced snow darkening effect (SDE), with a particular focus on the impacts on snow albedo and the surface energy balance. The snow depth data are employed only as an initial condition input to assess the influence of snow conditions on the SDE. Given that the previous title, "Modeling study of the snow darkening effect by black carbon deposition over the Arctic during the melting

period," did not accurately convey the main purpose of this article, we changed the title to "A numerical sensitivity study on the snow darkening effect by black carbon deposition over the Arctic in spring".

In this study, a fixed mixing ratio of BC was used to eliminate the influence of the BC concentration on the SDE. Previous observations of BC in Arctic snow indicate that the concentration of BC can vary from less than 5 to several hundred ng g$^{-1}$ in the spring. During the melting period, BC can accumulate on the snow surface because of its insolubility (Forsström et al., 2013). Therefore, the BC concentration in Arctic snow may be higher than the reported values from the initial melting period. Considering these factors and our previous study (Zhang et al., 2024), a mixing ratio of 50 ng g$^{-1}$ BC in Arctic snow was selected. This value is reasonable for assessing the potential impacts of snow processes and land–atmosphere interactions on the SDE, and its magnitude is realistic.

The simulated biases were caused mainly by the model resolution and inaccurate land surface characteristics used in this study. We added the relevant content to the introduction of the manuscript. In this study, simulation bias does not affect our main conclusions, as we are focused primarily on the impact of surface feedback mechanisms on BC-induced albedo changes. While the 2-m air temperature can influence surface energy exchange processes, the wind speed is not related to the mechanisms of interest in this paper. These biases do not affect the mechanisms we are focusing on.

The Polar-WRF model is a model specifically designed for polar regions, and its simulation capabilities have been validated in numerous studies (e.g., Hines & Bromwich, 2017; Pilguj et al., 2018; Turton et al., 2020). Comprehensive validation of the model's simulation capabilities is important, but it is not the focus of this paper. In the future, we will conduct more detailed studies on the reduction in snow albedo caused by BC and the corresponding changes in the surface energy balance in the Arctic region.

References

Hines, K. M., & Bromwich, D. H. Simulation of Late Summer Arctic Clouds during ASCOS with Polar WRF. Monthly Weather Review, 145(2), 521-541.·doi:10.1175/MWR-D-16-0079.1,2017

Pilguj, N., Czernecki, B., Kryza, M., Migała, K., & Kolendrowicz, L. Application of the Polar WRF model for Svalbard - sensitivity to planetary boundary layer, radiation and microphysics schemes. Polish Polar Research, 39.2018

Zou, X., Bromwich, D. H., Montenegro, A., Wang, S.-H., & Bai, L. Major surface melting over the Ross Ice Shelf part II: Surface energy balance. Quarterly Journal of the Royal Meteorological Society, 147(738), 2895-2916.·doi:10.1002/qj.4105,2021

Zhang, Z., Zhou, L., & Zhang, M. A progress review of black carbon deposition on Arctic snow and ice and its impact on climate change. Advances in Polar Science, 35(2), 178-191.·doi:10.12429/j.advps.2023.0024,2024

*Comment 2:*

*The manuscript states that weather models with high temporal and spatial resolution and conducting ensemble simulations are important in accurately assessing the impacts of the snow-darkening effect (SDE). However, these statements are not well justified. Also, this research uses a resolution of 27 km, which is not high.*

We sincerely appreciate the valuable comments. We apologize for the misunderstanding and misuse of the term 'ensemble simulation.' We removed the corresponding statements from the manuscript.

The snow albedo reduction caused by BC is directly related to the distribution of snow cover and terrain height. Kang et al. (2020) emphasized the importance of enhancing the spatial resolution of models to more effectively capture the spatial variability in snowpack. Therefore, it is essential to use high-resolution regional simulations that can better capture surface conditions. In this study, we used 0.25°×0.25° ERA5 reanalysis data as the initial and boundary conditions for the model. The 27 km resolution is consistent with the ERA5 reanalysis data to ensure the accuracy of large-scale meteorological conditions. In addition, previous studies have employed global models to evaluate the radiative forcing of BC in Arctic snow, typically with resolutions greater than 1° (e.g., Dong et al., 2018; Jiao et al., 2014; Ren et al., 2020). Thus, a resolution of 27 km has already demonstrated significant improvement. A higher-resolution modeling study will be conducted with the nesting method in the future.

References

Dou, T., Xiao, C., Shindell, D. T., Liu, J., Eleftheriadis, K., Ming, J., & Qin, D. The distribution of snow black carbon observed in the Arctic and compared to the GISS-PUCCINI model. Atmospheric Chemistry and Physics, 12(17), 7995-8007.·doi:10.5194/acp-12-7995-2012,2012

Jiao, C., Flanner, M. G., Balkanski, Y., Bauer, S. E., Bellouin, N., Berntsen, T. K., Bian, H., et al. An AeroCom assessment of black carbon in Arctic snow and sea ice. Atmospheric Chemistry and Physics, 14(5), 2399-2417.·doi:10.5194/acp-14-2399-2014,2014

Kang, S., Zhang, Y., Qian, Y., & Wang, H. A review of black carbon in snow and ice and its impact on the cryosphere. Earth-Science Reviews, 210.·doi:10.1016/j.earscirev.2020.103346,2020

Ren, L., Yang, Y., Wang, H., Zhang, R., Wang, P., & Liao, H. Source attribution of Arctic black

carbon and sulfate aerosols and associated Arctic surface warming during 1980–2018. Atmospheric Chemistry and Physics, 20(14), 9067-9085.·doi:10.5194/acp-20-9067-2020,2020

*Comment 3:*

*The manuscript needs a more comprehensive literature review. There are more studies focusing on BC impacts on snow surfaces using SNICAR over the Arctic areas.*

Thank you for the comment. The physically based Snow, Ice, Aerosol, and Radiation (SNICAR) model (Flanner et al., 2012; Flanner & Zender, 2005) is widely used to estimate the contributions of BC to the reduction in snow albedo and its corresponding radiative forcing (RF) in the Arctic. Flanner et al. (2007) incorporated SNICAR into the National Center for Atmospheric Research Community Atmosphere Model (NACR CAM3, Version 3) global climate model (GCM) to improve the quantification of climate forcing from BC in snow. Using the NCAR CAM5 coupled with SNICAR, Zhou et al. (2012) reported that in the spring season, the Arctic forcing increases from +0.29 W m$^{-2}$ to +0.37 W m$^{-2}$ due to BC deposition. On the basis of field observations conducted by Doherty et al. (2010), Dang et al. (2017) used SNICAR to calculate the reduction in snow albedo caused by BC in the Arctic and highlighted the impact of snowpack properties on the assessment of the SDE. In addition, SNICAR was also applied to quantify the reduction in snow and ice albedo caused by long-range-transported Asian dust (Zhao et al., 2022). All of the aforementioned studies were reviewed and added to the introduction. (L89-100)

References

Flanner, M. G., & Zender, C. S. Snowpack radiative heating: Influence on Tibetan Plateau climate. Geophysical Research Letters, 32(6).·doi:10.1029/2004GL022076,2005

Flanner, M. G., Zender, C. S., Randerson, J. T., & Rasch, P. J. Present-day climate forcing and response from black carbon in snow. Journal of Geophysical Research, 112(D11).·doi:10.1029/2006jd008003,2007

Flanner, M. G., Liu, X., Zhou, C., Penner, J. E., & Jiao, C. Enhanced solar energy absorption by internally-mixed black carbon in snow grains. Atmos. Chem. Phys., 12(10), 4699-4721.·doi:10.5194/acp-12-4699-2012,2012

Dang, C., Warren, S. G., Fu, Q., Doherty, S. J., Sturm, M., & Su, J. Measurements of light‐absorbing particles in snow across the Arctic, North America, and China: Effects on surface albedo. Journal of Geophysical Research: Atmospheres, 122(19). • doi:10.1002/2017jd027070,2017

Doherty, S. J., Warren, S. G., Grenfell, T. C., Clarke, A. D., & Brandt, R. E. Light-absorbing impurities in Arctic snow. Atmospheric Chemistry and Physics, 10(23), 11647-

11680.·doi:10.5194/acp-10-11647-2010,2010

Zhao, X., Huang, K., Fu, J. S., & Abdullaev, S. F. Long-range transport of Asian dust to the Arctic: identification of transport pathways, evolution of aerosol optical properties, and impact assessment on surface albedo changes. Atmos. Chem. Phys., 22(15), 10389-10407.·doi:10.5194/acp-22-10389-2022,2022

Zhou, C., Penner, J. E., Flanner, M. G., Bisiaux, M. M., Edwards, R., & McConnell, J. R. Transport of black carbon to polar regions: Sensitivity and forcing by black carbon. Geophysical Research Letters, 39(22).·doi:10.1029/2012GL053388,2012

*Comment 4:*

*The Mellor-Yamada-Nakanishi-Niino (MYNN) scheme is used to represent the boundary layer. However, modeling the Arctic boundary layer is very challenging. Please explain why the MYNN is selected and how it behaves over the Arctic. Also, do clouds play a role in the simulations?*

We sincerely appreciate the valuable comments. In this study, all the physical parameterization options applied in this study are based on a wide range of studies of PWRF over the Arctic (e.g, Hines & Bromwich, 2017; Wilson et al., 2011; Zou et al., 2021). Two PBL schemes are preferred when the PWRF model is used. The Mellor–Yamada–Janjić (MYJ) turbulent kinetic energy scheme (Janji, 1994) was first tested by the Polar Meteorology Group and has demonstrated superior performance on Arctic land and ocean surfaces (Bromwich et al., 2009; Wilson et al., 2011). However, the MYNN PBL scheme became more widely used after the WRFV4 release. The new version of the MYNN 2.5-level scheme implemented in WRF/PWRF Version 4.1.1 can improve downward shortwave radiation at the surface (Olson et al., 2019), and its performance was also validated by Zou et al. (2021) and Xue et al. (2021). In our study, downward shortwave radiation at the surface was a key factor in assessing the SDE caused by BC. Therefore, the MYNN scheme was chosen to represent the Arctic boundary layer. We also added the relevant description in L268-270.

In our study, the Morrison 2-moment scheme for cloud microphysics was used, which has also been widely tested in the Arctic (Hines & Bromwich, 2017; Hines et al., 2019; Turton et al., 2020; Xue et al., 2021). The effects of clouds, such as their impact on radiation and snowfall, have already been considered in the model. However, exploring the role of clouds is not the focus of this paper, and further research is needed in the future.

References
Bromwich, D. H., Hines, K. M., & Bai, L.-S. Development and testing of Polar Weather Research and Forecasting model: 2. Arctic Ocean. Journal of Geophysical Research: Atmospheres,

114(D8).·doi:10.1029/2008JD010300,2009

Hines, K. M., & Bromwich, D. H. Simulation of Late Summer Arctic Clouds during ASCOS with Polar WRF. Monthly Weather Review, 145(2), 521-541.·doi:10.1175/MWR-D-16-0079.1,2017

Janji, Z. I. The Step-Mountain Eta Coordinate Model: Further Developments of the Convection, Viscous Sublayer, and Turbulence Closure Schemes. Mon.wea.rev, 122(5), 927.·doi:10.1175/1520-0493(1994)122%3C0927:TSMECM%3E2.0.CO;2,1994

Olson, J. B., Kenyon, J. S., Angevine, W. A., Brown, J. M., Pagowski, M., & Sušelj, K. A Description of the MYNN-EDMF Scheme and the Coupling to Other Components in WRF–ARW [Technical Memorandum].·doi:10.25923/n9wm-be49,2019

Wilson, A. B., Bromwich, D. H., & Hines, K. M. Evaluation of Polar WRF forecasts on the Arctic System Reanalysis domain: Surface and upper air analysis. Journal of Geophysical Research: Atmospheres, 116(D11).·doi:10.1029/2010JD015013,2011

Xue, J., Bromwich, D. H., Xiao, Z., & Bai, L. Impacts of initial conditions and model configuration on simulations of polar lows near Svalbard using Polar WRF with 3DVAR. Quarterly Journal of the Royal Meteorological Society, 147(740), 3806-3834.·doi:10.1002/qj.4158,2021

Zou, X., Bromwich, D. H., Montenegro, A., Wang, S.-H., & Bai, L. Major surface melting over the Ross Ice Shelf part II: Surface energy balance. Quarterly Journal of the Royal Meteorological Society, 147(738), 2895-2916.·doi:10.1002/qj.4105,2021

*Comment 5:*

*Please explain the difference between CLM and Noah-MP, and highlight why Noah-MP is selected.*

We are grateful to the reviewer for these comments. In this study, the greatest difference between CLM and Niah-MP is their parameterizations for snow. The snowpack in the CLM can reach twelve layers (Lawrence et al., 2019), whereas that in the Noan-MP can reach three layers (Niu et al., 2011). Both models are commonly used land-surface schemes, and numerous studies have compared the differences between them (e.g., Chen et al., 2014; Constantinidou et al., 2020; Van Den Broeke et al., 2018). However, in this study, we evaluate the impact of BC on the Arctic surface energy balance via Polar-WRF, which is specifically optimized for simulating polar regions. In Polar-WRF, only the Noah and Noah-MP land surface models (LSMs) have been modified and optimized for polar regions, and Noah-MP performs better than Noah in simulating snow processes. Consequently, the Noah-MP model was selected for this study, and SNICAR was employed as the snow albedo scheme within the Polar-WRF/Noah-MP framework.

References

Chen, F., Liu, C., Dudhia, J., & Chen, M. A sensitivity study of high‑resolution regional climate simulations to three land surface models over the western United States. Journal of Geophysical Research: Atmospheres, 119, 7271 - 7291. • doi: 10.1002/2014JD021827,2014

Constantinidou, K., Hadjinicolaou, P., Zittis, G., & Lelieveld, J. Sensitivity of simulated climate over the MENA region related to different land surface schemes in the WRF model. Theoretical and Applied Climatology, 141(3), 1431-1449.·doi:10.1007/s00704-020-03258-5,2020

Lawrence, D. M., Fisher, R. A., Koven, C. D., Oleson, K. W., Swenson, S. C., Bonan, G., Collier, N., et al. The Community Land Model Version 5: Description of New Features, Benchmarking, and Impact of Forcing Uncertainty. Journal of Advances in Modeling Earth Systems, 11(12), 4245-4287.·doi:10.1029/2018MS001583,2019

Niu, G.-Y., Yang, Z.-L., Mitchell, K. E., Chen, F., Ek, M. B., Barlage, M., Kumar, A., et al. The community Noah land surface model with multiparameterization options (Noah-MP): 1. Model description and evaluation with local-scale measurements. Journal of Geophysical Research: Atmospheres, 116(D12).·doi:10.1029/2010JD015139,2011

Van Den Broeke, M. S., Kalin, A., Alavez, J. A. T., Oglesby, R., & Hu, Q. A warm-season comparison of WRF coupled to the CLM4.0, Noah-MP, and Bucket hydrology land surface schemes over the central USA. Theoretical and Applied Climatology, 134(3), 801-816.·doi:10.1007/s00704-017-2301-8,2018

*Comment 6:*

*A five-day spin up is used. However, it is unclear whether this is sufficient. For example, less variabilities are simulated during the first few days, as shown in the temporal evolution plots. Should these first few days also be considered spin-up?*

Thank you for your valuable comment. In this study, we employed the ERA5 reanalysis data for the initial and boundary conditions and enabled upper-air nudging to maintain consistency between the continuously updated boundary conditions and the initial conditions. Thus, the 5-day spin-up time is sufficient for atmospheric conditions (Gómez-Navarro et al., 2015; Jerez et al., 2020), but the surface requires more time to respond (Chen et al., 1997; Cosgrove et al., 2003).

In this study, the low variability observed in the temporal evolution plots can be attributed to changes in snowpack properties. As the temporal evolution plots indicate, the snow is fresh and has a low density during the initial days, which can diminish the changes in snow albedo caused by BC. In contrast, as snow ages and begins to melt, the effects of the snow darkening effect (SDE) induced

by BC increase. This evolution of snow properties and its impact on snow albedo reduction is one of the mechanisms that this study aims to highlight; therefore, we included it in this study.

References

Chen, F., Janjić, Z., & Mitchell, K. Impact of Atmospheric Surface-layer Parameterizations in the new Land-surface Scheme of the NCEP Mesoscale Eta Model. Boundary-Layer Meteorology, 85(3), 391-421.·doi:10.1023/A:1000531001463,1997

Cosgrove, B. A., Lohmann, D., Mitchell, K. E., Houser, P. R., Wood, E. F., Schaake, J. C., Robock, A., et al. Land surface model spin-up behavior in the North American Land Data Assimilation System (NLDAS). Journal of Geophysical Research: Atmospheres, 108(D22).·doi: 10.1029/2002JD003316,2003

Gómez-Navarro, J. J., Raible, C. C., & Dierer, S. Sensitivity of the WRF model to PBL parametrisations and nesting techniques: evaluation of wind storms over complex terrain. Geosci. Model Dev., 8(10), 3349-3363.·doi:10.5194/gmd-8-3349-2015,2015

Jerez, S., López-Romero, J. M., Turco, M., Lorente-Plazas, R., Gómez-Navarro, J. J., Jiménez-Guerrero, P., & Montávez, J. P. On the Spin-Up Period in WRF Simulations Over Europe: Trade-Offs Between Length and Seasonality. Journal of Advances in Modeling Earth Systems, 12(4), e2019MS001945.·doi:10.1029/2019MS001945,2020

*Comment 7:*

*Please provide more detailed description about experimental design. For example, which experiments are offline and which ones are online. Also, what's the meaning of "except for snow depth" etc.*

Thank you for your comments. In this study, the SNICAR-OFF and SEN experiments are offline simulations. SNICAR-OFF is an offline coupled simulation that is used to calculate the baseline of the SDE induced by BC deposition without snow cover changes or land–atmosphere exchange processes. The online coupled simulation (SNICAR-ON) is fully coupled with Polar-WRF, and the impacts of the SDE at every model timestep are computed by contrasting the dirty and clean snow albedos under the current surface and atmospheric conditions. Discrepancies between SNICAR-OFF and SNICAR-ON outcomes demonstrate the importance of atmospheric and surface feedback processes in comprehensively assessing the impacts of snow darkening effects (SDEs).

The SEN experiments are offline sensitivity experiments for the SNICAR model. SEN1-3 were designed to test the effects of snow density, snow depth, and snow grain size on the reduction in snow albedo caused by BC, respectively. SEN4 was designed to test the impact of the distribution

of BC within snow on BC-induced changes in snow albedo. The purpose of the SEN experiments is to test the sensitivity of SNICAR to snow properties and to preliminarily assess the impact of snow characteristics on the reduction in snow albedo caused by BC. 'Except for snow depth' means testing the impact of snow depth on the reduction in snow albedo caused by BC while keeping all other conditions the same. To improve understanding, we added the corresponding content to the manuscript (L294-315) and reorganized Table 1 as follows:

**Table 1. Summary of model simulations**

| Name | Mixing ratio of BC (ng g$^{-1}$) | Snow density (kg m$^{-3}$) | Snow depth (m) | Snow grain radius (μm) | BC distribution with snow | Surface feedback processes |
|---|---|---|---|---|---|---|
| CTL | 0 | Provided by NCEP GDAS final analysis data | Provided by NCEP GDAS final analysis data | 100 μm (new snow) or 1000 μm (old snow) | Vertically uniform distribution | Included |
| SNICAR-OFF | 50 | Same as CTL | Same as CTL | Same as CTL | Same as CTL | Not included |
| SNICAR-ON | 50 | Same as CTL | Same as CTL | Same as CTL | Same as CTL | Included |
| SEN1 | 50 | Same as CTL | Same as CTL | Set as 50, 150, 250,500,100 | Same as CTL | Not included |
| SEN2 | 50 | Set as 100, 200,500 | Same as CTL | Same as CTL | Same as CTL | Not included |
| SEN3 | 50 | Same as CTL | Set as 0.05, 0.1,0.25,0.5,1.0 | Same as CTL | Same as CTL | Not included |

| | | | | | | |
|---|---|---|---|---|---|---|
| SEN4 | 50 | Same as CTL | Same as CTL | Same as CTL | BC at top 5 cm layer | Not included |

*Comment 8:*

*Please clarify how the model accounts for snow-ice transition. For example, after melting and refreezing, can we still treat the surface as snow? As the study does not account for ice, clarification like this is necessary.*

Thank you for the valuable comment. In this work, we retained the treatment of snow hydrological processes and snow–ice transitions from the original Noah-MP model without making any modifications. In the Noah-MP model, snow consists of liquid water and ice particles. Initially, the liquid water content within the snowpack is zero. When the simulation begins, the snow ice content and liquid water content in each snow layer are updated whenever melting or refreezing occurs, provided that the snowpack has explicit snow layers (snow depth >= 2.5 cm). If snow is present but its thickness is insufficient to form an explicit snow layer, the snow ice content and liquid water content will no longer be calculated, and all liquid water in the snow is assumed to be ponded on the soil surface (He et al., 2023). The liquid water content and ice content in each snow layer affect the total column snow mass and snow optical thickness, which consequently influence the reduction in snow albedo caused by BC. More detailed descriptions are provided in He et al. (2023).

References

He, C., P. V., M. Barlage, F., Chen, D., Gochis, R., Cabell, T., Schneider, R. R., et al. The Community Noah-MP Land Surface Modeling System Technical Description Version 5.0, (No. NCAR/TN-575+STR).·doi:10.5065/ew8g-yr95,2023

*Comment 9:*

*Section 3.4 is hard to follow. The time series plots show dense information, but the explanations are brief. It is unclear how shallow, moderate, and deep snow depths are defined. There are brief explanations on Line 444, yet information like this should be placed in the beginning of the section and also be expanded on to give a clear background to facilitate a better understanding of the temporal evolutions.*

Thank you for the valuable comment. In this study, the three snow conditions are determined on the basis of the total snow depth ($h_{sno}$): shallow ($0.025m < h_{sno} < 0.25m$), moderate

( $0.25m \leq h_{sno} < 0.45m$ ) and deep ( $h_{sno} \geq 0.45m$ ). The detailed information is shown in Appendix A3.

For clarity, we adjusted the order and added more detailed explanations at the beginning of Section 3.4 as follows: (L460-467).

"As discussed above, the impacts of the SDE can be affected by changes in snowpack properties through two primary mechanisms. First, this study assumed that the distribution of BC was uniform throughout the snowpack, indicating that deeper layers of snow contained higher concentrations of BC particles. The overall mass of BC present in snow can influence the modeling effects of the SDE. In addition, as snow ages and begins to melt, the effects of the SDE induced by BC increase due to the reduced snow albedo (He & Ming, 2022). To analyze the impact of the evolution of snow on the SDE, this study categorized snow into three types on the basis of snow depth and investigated the related changes on the SDE as snow ages and melts. The three snow conditions are determined on the basis of the total snow depth ($h_{sno}$): shallow ($0.025m < h_{sno} < 0.25m$), moderate ($0.25m \leq h_{sno} < 0.45m$) and deep ($h_{sno} \geq 0.45m$). The detailed information is shown in Appendix A3."

References

He, C., & Ming, J. Modelling light-absorbing particle–snow–radiation interactions and impacts on snow albedo: fundamentals, recent advances and future directions. Environmental Chemistry, 19(5), 296-311.·doi:10.1071/en22013,2022

*Minor comments:*

*Comment 1:*

*Inconsistent titles in the manuscript and the supplementary materials.*

We apologize for our carelessness and mistakes. We revised the title of the supplementary material and made it more consistent with the manuscript.

*Comment 2:*

*In the result section, please clarify which results are coming from which experiments.*

Thank you for the comment. In this study, the RFs caused by BC were calculated via both SNICAR-ON and SNICAR-OFF. The relationship between snow depth and snow albedo reduction by BC was observed in SNICAR-OFF. The temporal evolution of the changes in the surface energy balance was investigated via SNICAR-ON. The importance of snow melting and the land −

atmosphere exchange process was evaluated by comparing the results from online (SNICAR-ON) and offline (SNICAR-OFF) coupled simulations. The relevant content was included in the Results section.

*Comment 3:*

*Many typos, such as Line 105 "through and", Line 318 "f-j", Line 342 "each maximum levels"; Line 481 "shows the how," etc.*

We apologize for our carelessness and mistakes. We checked and revised all the typos that we could find.

*Comment 4:*

*Please add more descriptions regarding the advantage of Polar-WRF. Simply refereeing to other documents is not good enough to provide a background for this study.*

Thank you for the comments. The main advantages of the Polar-WRF model include (a) optimizing the treatment of heat transfer for ice sheets and revised surface energy balance calculations in the Noah and Noah-MP LSMs; (b) comprehensively describing sea ice in Noah and Noah-MP; and (c) improving cloud microphysics for polar regions. Detailed descriptions of Polar-WRF were added to the manuscript (L146-160).

*Comment 5:*

*Please provide brief description of the data downloaded from the Arctic Data Center.*

Thank you for the comment. The Arctic Data Center is the primary data and software repository for the Arctic section of the National Science Foundation's Office of Polar Programs. In this study, we downloaded the energy exchange data measured by an eddy covariance system in terrestrial systems across Alaska. We also added a brief description of the data in the manuscript as follows (L280-283).

"The observed downward shortwave radiation, sensible heat flux and latent heat flux in Alaska (149.3°W, 68.6°N) were downloaded from the Arctic Data Center (https://arcticdata.io/). These measurements were conducted by the University of Alaska Fairbanks (UAF) and are part of the Arctic Observing Network (AON) project. For these measurements, an eddy covariance system was employed to measure the fluxes of $CO_2$, water, and energy, and it was positioned on a 3-meter-high tripod in the center of the site. The data were averaged over 30-minute intervals and span the entire

year, with quality control measures implemented.".

*Comment 6:*

*Line 325, how is the "absolute mass of BC represented by the snow depth"?*

Thank you for the comment. In this study, we assume that the distribution of BC is uniform throughout the snowpack, indicating that deeper layers of snow contain a greater mass of BC particles. To make this easier to understand, we revised it to "the mass of BC in snow is directly related to snow depth."

*Comment 7:*

*For Section 2.3, please clarify whether the calculations are included in WRF as one of the diagnostics, or are these offline calculations based on modeled results.*

Thank you for the comment. In this study, the calculations of the surface energy balance were coupled into the Polar-WRF. We clarified this in the manuscript (L229).

*Comment 8:*

*Please clarify "The high RF caused by BC deposition reached 1-4 W m^-2"*

Thank you for the comment. In this study, the most pronounced RF caused by BC occurred mainly in regions with deep snow depths and high snow densities. The RF caused by BC deposition can reach greater than 4 W m$^{-2}$ in Greenland, Baffin Island, and East Siberia. We modified this sentence in the manuscript for clarity (L620-622).

---

## Author Response (AR2)

*The authors provided reasonable explanations for most of the questions. A few clarifications still needed:*

We appreciate that the editor and reviewers recognize our efforts and thank you for your thoughtful suggestions and insights, which have helped improve this manuscript substantially. We have revised the manuscript carefully, as described in our point-to-point responses to the comments.

*1. About authors' response "The simulated biases were caused mainly by the model resolution and inaccurate land surface characteristics used in this study," please clarify "inaccurate land surface characteristics" and whether this may influence understanding of surface processes, which is the focus of this study.*

Thank you for the comment. In this study, we focused on two surface physical processes: the aging and melting processes of snowpack and the land–atmosphere exchange process. These processes have been integrated into the Polar-WRF/Noah-MP model, which has been extensively validated (e.g., Justino et al., 2019; Li et al., 2022; Smith et al., 2017). The inaccuracy of surface characteristics in this study is mainly due to the land cover and topography inputs, which can introduce biases in the findings. Nevertheless, this does not impact the surface processes and physical mechanisms that were the main focus of the study. As suggested, we clarified it and added explanation in the paper (P10, L339-342 and L348) as follows:

"Surface feedback processes such as land–atmosphere exchange and changes of snowpack have already coupled into the Polar-WRF/Noah-MP and their performances have been widely validated (e.g., Justino et al., 2019; Li et al., 2022; Smith et al., 2017). To better access the impacts of these two surface feedback processes on the reduction in snow albedo caused by BC deposition and the corresponding changes in the surface energy balance, the modeled downward shortwave radiation, sensible heat flux (HS) and latent heat flux (LH) are also compared with the observation data in Alaska (149.3°W, 68.6°N)."

"These biases may result from the inaccurate land surface characteristics (e.g., land cover and topography) used in this study and the coarse model resolution."

References

Li, J., Miao, C., Zhang, G., Fang, Y.-H., Shangguan, W., & Niu, G.-Y. Global Evaluation of the Noah-MP Land Surface Model and Suggestions for Selecting Parameterization Schemes. Journal of Geophysical Research: Atmospheres, 127(5), e2021JD035753.·doi:10.1029/2021JD035753,2022

Justino, F., Wilson, A. B., Bromwich, D. H., Avila, A., Bai, L.-S., & Wang, S.-H. Northern Hemisphere Extratropical Turbulent Heat Fluxes in ASRv2 and Global Reanalyses. Journal of Climate, 32(7), 2145-2166.·doi:10.1175/JCLI-D-18-0535.1,2019

Smith, W. L., Hansen, C., Bucholtz, A., Anderson, B., Beckley, M., Corbett, J. G., Cullather, R. I., et al. Arctic Radiation-IceBridge Sea and Ice Experiment: The Arctic Radiant Energy System during the Critical Seasonal Ice Transition. Bulletin of the American Meteorological Society, 98, 1399-1426.2017

2. *If "the wind speed is not related to the mechanisms of interest in this paper," please clarify the rationale of including this comparison/validation in the paper.*

*Similarly, as this paper primarily focused on surface processes, many descriptions related to atmospheric setups (e.g., boundary layer, clouds, etc) may appear redundant. However, I believe these processes are crucial for accurately simulating precipitation and snowfall. Therefore, selecting a better PBL scheme and cloud mechanism is important and should be emphasized in the paper.*

Thank you for the comment. In this study, we focused on BC-induced reduction in snow albedo and their associated surface feedback processes (the changes of snowpack properties and land–atmosphere exchange), the role of wind speed was not emphasized. However, the wind speed is still one of the most important meteorological parameters, which is closely related to the surface energy balance and the structure of the atmospheric boundary layer. Thus, the comparison is necessary to validate the basic capabilities of the model and to conduct further investigations.

The Mllor-Yamada-Nakanishi-Niino (MYNN) level 2.5 PBL scheme and the Morrison 2-moment cloud microphysics scheme were selected in this study. Their performances in the Arctic are widely have been widely tested and verified (e.g., Hines & Bromwich, 2017; Hines et al., 2019; Turton et al., 2020; Xue et al., 2021). As suggested, we added more detailed description of these schemes and emphasized their importance in the paper (P7-8, L260-276):

"In addition to surface processes, atmospheric conditions like the boundary layer and clouds play a key role in effectively simulating precipitation and snowfall, which can influence the reliability of the simulation outcomes. As a result, choosing the appropriate boundary layer and cloud microphysics schemes is essential. In this study, the Mllor-Yamada-Nakanishi-Niino (MYNN) level 2.5 PBL scheme and the Morrison 2-moment cloud microphysics scheme were selected. Their

performances in the Arctic are widely have been widely tested and verified (e.g., Hines & Bromwich, 2017; Hines et al., 2019; Turton et al., 2020; Xue et al., 2021).The MYNN model is a kind of second-order closure model that was proposed by Nakanishi and Niino (Nakanishi and Niino 2004, 2006, 2009) and is formulated as a modification of the Mellor-Yamada closure model (Mellor and Yamada 1982). In comparison to the MYNN level-3 scheme, the MYNN 2.5-level scheme retains the significant performance on the stable boundary layer simulations and reduces the computational cost (Kitamura, 2010; Nakanishi & Niino, 2009). The new version of the MYNN 2.5-level scheme implemented in WRF/PWRF Version 4.1.1 can improve downward shortwave radiation at the surface (Olson et al., 2019), which is a key factor in assessing the reduction in snow albedo caused by BC deposition and the corresponding changes in the surface energy balance.

The Morrison 2-moment cloud microphysics scheme is a double-moment microphysics scheme that parameterizes the mixing ratio and number concentration of hydrometeors, covering cloud droplets, rain, ice crystals, snow, and graupel (Morrison & Gettelman, 2008). In the Polar-WRF, its droplet concentration is reduced from 250 $cm^{-3}$ to 50 $cm^{-3}$, which is more applicable to polar regions (Hines & Bromwich, 2017). It has been extensively tested and has shown a great simulation capabilities, especially in the representation of mixed-phase clouds in the Arctic (Arteaga et al., 2024; Cho et al., 2020)."

References

Arteaga, D., Planche, C., Tridon, F., Dupuy, R., Baudoux, A., Banson, S., Baray, J.-L., et al. Arctic mixed-phase clouds simulated by the WRF model: Comparisons with ACLOUD radar and in situ airborne observations and sensitivity of microphysics properties. Atmospheric Research, 307, 107471.·doi:10.1016/j.atmosres.2024.107471,2024

Cho, H., Jun, S.-Y., Ho, C.-H., & McFarquhar, G. Simulations of Winter Arctic Clouds and Associated Radiation Fluxes Using Different Cloud Microphysics Schemes in the Polar WRF: Comparisons With CloudSat, CALIPSO, and CERES. Journal of Geophysical Research: Atmospheres, 125(2), e2019JD031413.·doi:10.1029/2019JD031413,2020

Hines, K. M., & Bromwich, D. H. Simulation of Late Summer Arctic Clouds during ASCOS with Polar WRF. Monthly Weather Review, 145(2), 521-541.·doi:10.1175/MWR-D-16-0079.1,2017

Kitamura, Y. Modifications to the Mellor-Yamada-Nakanishi-Niino (MYNN) Model for the Stable Stratification Case. Journal of the Meteorological Society of Japan. Ser. II, 88(5), 857-864.·doi:10.2151/jmsj.2010-506,2010

Mellor, G. L., & Yamada, T. Development of a turbulence closure model for geophysical fluid problems. Reviews of Geophysics, 20(4), 851-875.·doi:10.1029/RG020i004p00851,1982

Morrison, H., & Gettelman, A. A New Two-Moment Bulk Stratiform Cloud Microphysics Scheme in the Community Atmosphere Model, Version 3 (CAM3). Part I: Description and Numerical Tests. Journal of Climate, 21(15), 3642-3659.·doi:10.1175/2008JCLI2105.1,2008

Nakanishi, M., & Niino, H. Development of an Improved Turbulence Closure Model for the Atmospheric Boundary Layer. Journal of the Meteorological Society of Japan. Ser. II, 87(5), 895-912.·doi:10.2151/jmsj.87.895,2009

Olson, J. B., Kenyon, J. S., Angevine, W. A., Brown, J. M., Pagowski, M., & Sušelj, K. A Description of the MYNN-EDMF Scheme and the Coupling to Other Components in WRF–ARW [Technical Memorandum].·doi:10.25923/n9wm-be49,2019

*3. 27 km is higher than typical resolutions used in global simulations. 27 km cannot be considered a high resolution by itself. Please make this clear in the paper.*

Thank you for the comment. As suggested, we included relevant sentences to clarify this as follows (P7, L242-245):

"The 27 km resolution is consistent with the ERA5 reanalysis data to ensure the accuracy of large-scale meteorological conditions, and it is significantly higher than the usual resolution employed in global climate models (which is typically over 1°) in earlier research (e.g., Dou et al., 2012; Jiao et al., 2014; Ren et al., 2020)."

References

Dou, T., Xiao, C., Shindell, D. T., Liu, J., Eleftheriadis, K., Ming, J., & Qin, D. The distribution of snow black carbon observed in the Arctic and compared to the GISS-PUCCINI model. Atmospheric Chemistry and Physics, 12(17), 7995-8007.·doi:10.5194/acp-12-7995-2012,2012

Jiao, C., Flanner, M. G., Balkanski, Y., Bauer, S. E., Bellouin, N., Berntsen, T. K., Bian, H., et al. An AeroCom assessment of black carbon in Arctic snow and sea ice. Atmospheric Chemistry and Physics, 14(5), 2399-2417.·doi:10.5194/acp-14-2399-2014,2014

Ren, L., Yang, Y., Wang, H., Zhang, R., Wang, P., & Liao, H. Source attribution of Arctic black carbon and sulfate aerosols and associated Arctic surface warming during 1980–2018. Atmospheric Chemistry and Physics, 20(14), 9067-9085.·doi:10.5194/acp-20-9067-2020,2020

*4. The discussion of how the model accounts for snow-ice transitions needs to be added to the paper, besides referring to He et al. (2023). This information is crucial to support readers' understanding about the processes.*

Thank you for the comment. As suggested, we added more relevant description about the snow process in the paper as follows (P6-7, L216-239):

"In the Noah-MP, the evolution of snowpack properties, including snow ice and liquid water content, snow thickness, and water flux out of snowpack bottom. If the snow layer temperature is higher than freezing point (273.15 K), then the snow layer ice is melting; if snow layer liquid water content is greater than 0, and snow layer temperature is lower than freezing point, then ice is refreezing. Once melting or freezing active, the snow ice amount will be updated. The amount of phase-change water is computed as:

$$\Delta W_{phase}(i) = \frac{H_{M,phase}(i) \times \Delta t}{C_{LH,fus}} \tag{7}$$

where $\Delta W_{phase}$ (kg m⁻²) is amount of phase-change water, $i$ is the snow layer, $H_{M,phase}$ (W m⁻²) is the energy residual (surplus or loss), and it is computed as:

$$H_{M,phase}(i) = \frac{T_{snso}(i) - T_{frz}}{\Delta t} \times C_{h,snow} \times \Delta z \tag{8}$$

where $T_{sno}$ (K) is the snow temperature, $T_{frz} = 273.15$ (K) is the freezing point, $\Delta z$ (m) is the thickness of snow layer, $C_{h,snow}$ (J m⁻³ K⁻¹) is the volumetric specific heat capacity of snow and it is calculated as:

$$C_{h,snow} = C_{h,ice} \times \theta_{ice,sno} + C_{h,wat} \times \theta_{liq,sno} \tag{9}$$

where $C_{h,ice}$ (J m⁻³ K⁻¹) and $C_{h,wat}$ (J m⁻³ K⁻¹) are the volumetric specific heat capacity of ice and water, respectively, $\theta_{ice,sno}$ and $\theta_{liq,sno}$ are partial volume of ice and liquid water in snow layer, respectively.

For each snow layer, if the freezing is active, then the snow ice content ($W_{ice,sno}$, [kg m⁻²]) is updated as:

$$W_{ice,sno,new}(i) = \min (W_{snow,old}(i), W_{ice,sno,old}(i) \tag{10}$$
$$- \Delta W_{phase}(i))$$

If the melting is active, then the snow ice content ($W_{ice,sno}$, [kg m⁻²]) is updated as:

$$W_{ice,sno,new}(i) = \max (0, W_{ice,sno,old}(i) - \Delta W_{phase}(i)) \tag{11}$$

Then, the snow liquid water content ($W_{liq,sno}$, [kg m⁻²] is updated as:

$$W_{liq,sno,new}(i) = \max (0, W_{snow,old}(i) - W_{ice,sno,new}) \tag{12}$$

As the snow melts, the amount of liquid water content in the snowpack will rise, leading to an increase in snow density. Once the liquid water content surpasses the snowpack's maximum capacity to hold water, the snowpack will start to flow out, resulting in a reduction in snow depth. These changes in snow properties will influence the snow albedo reduction caused by BC."

*5. My original comment "In the result section, please clarify which results are coming from*

*which experiments" was not fully addressed. There was a list of simulations performed, as shown in Table 1. Besides stating "SNICAR-ON" vs. "SNICAR-OFF" the authors also need to clarify each result section corresponds to which simulation setups.*

Thank you for the comment. We apologize for our carelessness. As recommended, we have revised and clarified the simulation setups in each results section. For example, in the Section 3.4, we clarified that the changes in the surface energy balance was investigated via the SNICAR-ON and the snow processes are included in this experiment (P15, L469-470):

"Based on SNICAR-ON simulation results (include the snow processes), the temporal evolution of the SDE caused by a fixed 50 ng $g^{-1}$ BC has been studied."